# Ab Initio Nonparametric Variable Selection for Scalable Symbolic Regression with Large $p$

**Shengbin Ye** [1 2]  **Meng Li** [1]

## Abstract

Symbolic regression (SR) is a powerful technique for discovering symbolic expressions that characterize nonlinear relationships in data, gaining increasing attention for its interpretability, compactness, and robustness. However, existing SR methods do not scale to datasets with a large number of input variables (referred to as extreme-scale SR), which is common in modern scientific applications. This "large $p$" setting, often accompanied by measurement error, leads to slow performance of SR methods and overly complex expressions that are difficult to interpret. To address this scalability challenge, we propose a method called PAN+SR, which combines a key idea of ab initio nonparametric variable selection with SR to efficiently pre-screen large input spaces and reduce search complexity while maintaining accuracy. The use of nonparametric methods eliminates model misspecification, supporting a strategy called parametric-assisted nonparametric (PAN). We also extend SRBench, an open-source benchmarking platform, by incorporating high-dimensional regression problems with various signal-to-noise ratios. Our results demonstrate that PAN+SR consistently enhances the performance of 19 contemporary SR methods, enabling several to achieve state-of-the-art performance on these challenging datasets.

## 1. Introduction

Symbolic regression (SR) is a mathematical technique for finding a symbolic expression that matches data from an unknown function. An early example of SR dates back to the 1600s when Johannes Kepler used astronomical data to discover that Mars' orbit was elliptical. This discovery, along with Kepler's other parsimonious and analytically tractable laws of planetary motion, helped launch a scientific revolution.

With the recent progress in theoretical modeling and experimental instrumentation, researchers have entered a new era of big data. The development of SR models is particularly important, as they have emerged as a powerful tool for developing machine learning models that are intelligible, interpretable, and compact. Unlike large numerical models, the mathematical expressions used in SR models enable an easy understanding of their behavior, making them valuable in fields such as physics, where they can connect newly discovered physical laws with theory to facilitate subsequent theoretical developments (Wu & Tegmark, 2019). Moreover, SR models offer a safe and responsible option for machine learning applications with high societal stakes, such as those related to human lives, as they are well-suited for human interpretability and in-depth analysis. As such, SR models have found successful applications across a range of fields, including astrophysics (Lemos et al., 2023), chemistry and materials science (Hernandez et al., 2019; Liu et al., 2020; 2022), control (Derner et al., 2020), economics (Verstyuk & Douglas, 2022), mechanical engineering (Kronberger et al., 2018), medicine (Virgolin et al., 2020), and space exploration (Märtens & Izzo, 2022), among others (Matsubara et al., 2024).

SR literature has traditionally focused on datasets with low-dimensional inputs, often with $p \leq 10$, and primarily considered only relevant variables—those used in the ground truth (La Cava et al., 2021; Kamienny et al., 2022; Shojaee et al., 2023; Tenachi et al., 2023; Li et al., 2024). In these settings, variable selection has not been critical, as SR has largely been viewed as an optimization problem under low-noise conditions. However, modern scientific applications increasingly involve datasets with far larger numbers of variables ($p = 102$ to $459$ in this work), often including irrelevant variables, rendering variable selection a critical yet underexplored concept in SR pipelines.

While variable selection is a well-established topic in statistics, its adoption in SR has been limited and its effective-

---

[1]Department of Statistics, Rice University, Houston, TX, USA [2]Department of Statistics and Data Science, Northwestern University, Evanston, IL, USA. Correspondence to: Meng Li <meng@rice.edu>.

*Proceedings of the 42nd International Conference on Machine Learning*, Vancouver, Canada. PMLR 267, 2025. Copyright 2025 by the author(s).

ness in SR remains unclear. Existing approaches, such as random forest (RF)-based pre-selection in PySR (Cranmer, 2023), have demonstrated limited utility. Indeed, the PySR documentation explicitly notes that options like `select_k_features` are rarely used, suggesting that current methods are not well-suited to SR tasks. This observation is further supported by our analysis in Appendix D.2, where RF is shown to perform unsatisfactorily. The limited performance of off-the-shelf methods like RF highlights the unique challenges of variable selection in the context of SR. Unlike typical variable selection tasks, SR variable selection demands a near-zero false negative rate (FNR), as excluding even a single relevant variable from the search space prevents the recovery of the true underlying function. While false positives (FPs) primarily increase computational burden, they do not fundamentally impede the discovery of the underlying model. This asymmetry in performance requirements explains why standard methods often fall short and underscores the importance of designing variable selection methods specifically tailored to SR.

In this paper, we introduce a versatile framework, PAN+SR, for improving SR methods at extreme scales. PAN+SR leverages the Parametric Assisted by Nonparametrics (PAN) strategy (Ye et al., 2024) for an *ab initio* screening of large influx of input variables before expression synthesis, enabling SR tasks at extreme scales. In light of the unique challenge of SR pre-screening, we propose a novel nonparametric variable selection method designed to minimize FN; we refer to this method as PAN throughout this paper. Furthermore, to evaluate PAN+SR at extreme scales, we extend the open-source SR benchmarking database, SRBench (La Cava et al., 2021), with high-dimensional problems containing white noise at various signal-to-noise ratios. In Section 6, we showcase the performance uplift of 19 contemporary SR methods under PAN+SR. The PAN+SR framework is available as an open-source project at https://github.com/mattsheng/PAN_SR.

## 2. Background and Motivation

Given a dataset $(\boldsymbol{y}, \boldsymbol{X})$ with target $\boldsymbol{y} \in \mathbb{R}^n$ and features $\boldsymbol{X} = (\boldsymbol{x}_1, \ldots, \boldsymbol{x}_p) \in \mathbb{R}^{n \times p}$, SR assumes the existence of an analytical data-generating function that links $\boldsymbol{X}$ to $\boldsymbol{y}$:

$$y_i = f_0(x_{i1}, \ldots, x_{ip}) + \varepsilon_i, \quad \text{for} \quad i = 1, \ldots, n, \quad (1)$$

in the presence of observation noise $\varepsilon_i$. The goal of SR is to recover the unknown regression function $f_0(\cdot)$ symbolically. For example, consider regressing the gravitational force between two objects, $F$, on their masses $(m_1, m_2)$ and the distance between their centers $(r)$. An SR algorithm would ideally re-discover the Newton's Law of Universal Gravitation, $F = 6.6743 \times 10^{-11} \cdot m_1 m_2 / r^2$. This is typically done by randomly constructing mathematical expressions using the

features, $\boldsymbol{X} = (m_1, m_2, r)$ in this case, and a set of mathematical operations, e.g., $\mathcal{O} = \{+, -, \times, \div, \exp, \log, \cdot^2\}$. Even for this low-dimensional problem, it has been shown that exploring all expressions $\mathcal{F}(\boldsymbol{X}, \mathcal{O})$, induced by $\boldsymbol{X}$ and $\mathcal{O}$, is NP-hard (Virgolin & Pissis, 2022). Hence, typical SR algorithms only traverse through a small subset of the full search space, such as limiting the complexity of the candidate SR models, total runtime, number of mathematical operations, etc.

In realistic scientific applications, particularly in the era of big data, scientists often include as many intuitively reasonable features as possible, many of which may be irrelevant to the target $\boldsymbol{y}$. This practice causes the search space $\mathcal{F}(\boldsymbol{X}, \mathcal{O})$ to expand double-exponentially quick (Ye et al., 2024), making it extremely challenging–if not impossible–to recover $f_0(\cdot)$ using algorithmic approaches alone. To this end, we propose the PAN+SR framework, which integrates the nonparametric module of PAN as a model-based pre-screening step. This framework excludes irrelevant features prior to applying SR methods, thereby mitigating the explosion of the search space in high-dimensional problems. Here, we assume that a high-dimensional SR problem in (1) can be reduced to

$$y_i = f_0(\boldsymbol{X}_{i, \mathcal{S}_0}) + \varepsilon_i, \quad \text{for} \quad i = 1, \ldots, n, \quad (2)$$

where only a small subset $\mathcal{S}_0$ of $p_0 = |\mathcal{S}_0| \ll p$ of features exert influence on $\boldsymbol{y}$. Then the oracle search space $\mathcal{F}(\boldsymbol{X}_{\mathcal{S}_0}, \mathcal{O})$ is a significantly smaller subspace of the full search space $\mathcal{F}(\boldsymbol{X}, \mathcal{O})$. Thus, the successful identification of $\mathcal{S}_0$, or at least a superset of $\mathcal{S}_0$, is critical for reducing high-dimensional SR problems into manageable low-dimensional ones. With this reduction, the dataset $(\boldsymbol{y}, \boldsymbol{X}_{\mathcal{S}_0})$ becomes sufficient for discovering $f_0(\cdot)$, enabling SR methods to handle high-dimensional problems without requiring any modifications to their algorithms.

## 3. Related Work

SRBench (La Cava et al., 2021) is a reproducible and open-source benchmarking platform for SR that has made significant strides in the field through its curation of 122 real-world datasets and 130 ground-truth problems and its comprehensive evaluations of 14 contemporary SR methods. SRBench has quickly gained adaptions with numerous studies leveraging it to evaluate accuracy, exact solution rate, and solution complexity (Kamienny et al., 2022; Landajuela et al., 2022; Kamienny et al., 2023; Keren et al., 2023; Shojaee et al., 2023; Makke & Chawla, 2024). Despite its widespread use, SRBench primarily focuses on low-dimensional problems, which limits its applicability in the context of high-dimensional problems, a hallmark of the era of big data. In particular, the 130 ground-truth problems from the Feynman Symbolic Regression Database (Udrescu & Tegmark, 2020)

and the ODE-Strogatz repository (Strogatz, 2015) contain only the oracle features $\boldsymbol{X}_{\mathcal{S}_0}$ with at most $p = 9$ features. This low and narrow dimensional scope leaves SRBench less suited for analyzing SR at extreme scales, underscoring the need for a high-dimensional SR database.

## 4. Method

Inspired by PAN, the PAN+SR framework utilizes a one-step nonparametric variable selection strategy to pre-screen a high-dimensional dataset $(\boldsymbol{y}, \boldsymbol{X})$ and parse the reduced dataset $(\boldsymbol{y}, \boldsymbol{X}_{\widehat{\mathcal{S}}})$ to SR methods for subsequent expression synthesis and selection. Unlike traditional variable selection literature, where the primary focus is controlling the false discovery rates, the PAN criterion calls for minimizing the false negative rate (FNR) while controlling the false positive rate (FPR) is secondary. In other words, the selected set of features $\widehat{\mathcal{S}}$ should be a superset of $\mathcal{S}_0$ and as small as possible. When $\widehat{\mathcal{S}}$ fails to be the superset of $\mathcal{S}_0$ (i.e., there is at least one FN), the reduced search space $\mathcal{F}(\boldsymbol{X}_{\widehat{\mathcal{S}}}, \mathcal{O})$ no longer contains $f_0(\cdot)$, rendering any subsequent discovery based on $\boldsymbol{X}_{\widehat{\mathcal{S}}}$ to be false.

Nonparametric or model-free variable selection has been extensively studied in the literature. Lafferty and Wasserman (2008) propose the RODEO method for nonparametric variable selection through regularization of the derivative expectation operator. Candès et al. (2018) propose a model-free knockoff procedure controlling FDR with no assumptions on the conditional distribution of the response. Fan et al. (2011) propose a sure independence screening method for B-spline additive model. In the Bayesian literature, Bleich et al. (2014) design permutation tests for variable inclusion proportion of Bayesian Additive Regression Tree (BART); Liu et al. (2021) deploy spike-and-slab priors directly on the nodes of Bayesian forests.

Despite this diverse array of methods, few meet the unique proposition of the PAN criterion. Among the few recent methods investigated in Ye et al. (2024), they found BART-G.SE (Bleich et al., 2014), a BART-based permutation variable selection method, to be particularly suitable for PAN. However, our comprehensive simulation study in Appendix D.2 reveals that BART-G.SE, along with three other methods, exhibit insufficient TPR, particularly under noisy or low-sample-size conditions. This deficiency renders these methods unsuitable for the PAN+SR framework.

In this paper, we introduce a novel BART-based variable selection method and demonstrate its PAN criterion consistency through an extensive simulation study in Section 6.2. The key idea behind BART is to model the regression func-tion $f_0(\cdot)$ by a sum of regression trees,

$$\boldsymbol{y} = \sum_{i=1}^{M} \mathcal{T}_i(\boldsymbol{x}_1, \ldots, \boldsymbol{x}_p) + \boldsymbol{\varepsilon}, \quad \boldsymbol{\varepsilon} \sim \mathcal{N}_n(\boldsymbol{0}, \sigma^2 \boldsymbol{I}_n), \quad (3)$$

where each regression tree $\mathcal{T}_i(\boldsymbol{x}_1, \ldots, \boldsymbol{x}_p)$ partitions the feature space based on the values of $\boldsymbol{x}_1, \ldots, \boldsymbol{x}_p$. For each posterior sample, we calculate the proportion of splits in the ensemble (3) that use $\boldsymbol{x}_j$ as the splitting variable, for $j = 1, \ldots, p$. The variable inclusion proportion (VIP) $q_j$ of $\boldsymbol{x}_j$ is then estimated as the posterior mean of these proportions across all posterior samples (Chipman et al., 2010). Intuitively, $q_1, \ldots, q_p$ encode the relative importance of each feature, where a large VIP $q_j$ suggests $\boldsymbol{x}_j$ being an important driver of the response $\boldsymbol{y}$. However, deciding on how large a VIP value must be to indicate relevance remains a challenge. For instance, BART-G.SE addresses this by using a permutation test on $q_1, \ldots, q_p$ to identify significant features, whereby controlling the family-wise error rate.

Here, we propose an alternative approach that utilizes the rankings of VIPs instead of their raw values. Specifically, let $r_j$ denote the ranking of the VIP $q_j$. Relevant features $\boldsymbol{X}_{\mathcal{S}_0}$ are expected to occupy top-ranking positions, namely $\{1, \ldots, p_0\}$, due to their strong associations with $\boldsymbol{y}$. In contrast, irrelevant features $\boldsymbol{X}_{\mathcal{S}_1}$, $\mathcal{S}_1 = [p] \setminus \mathcal{S}_0$, are expected to appear in lower-ranking positions, namely $\{p_0 + 1, \ldots, p\}$, since they are only selected sporadically or by chance (Chipman et al., 2010; Bleich et al., 2014). Consequently, a natural decision rule is to select feature $\boldsymbol{x}_j$ if $r_j$ falls within $\{1, \ldots, p_0\}$.

However, this decision rule is impractical in real-world applications since the sparsity $p_0$ is unknown. To address this limitation, we propose a method that leverages multiple independent runs of BART to estimate the feature rankings more robustly. Let $r_{j,k}$ denote the VIP ranking of $\boldsymbol{x}_j$ in the $k$th run. Assume that the rankings of $\boldsymbol{x}_j$ are randomly distributed over the $K$ independent runs (see Appendix D.1 for empirical justification):

$$r_{j,1}, \ldots, r_{j,K} \overset{\text{iid}}{\sim} \begin{cases} \text{Unif}(\{1, \ldots, p_0\}), & \text{if } j \in \mathcal{S}_0 \\ \text{Unif}(\{p_0 + 1, \ldots, p\}), & \text{if } j \notin \mathcal{S}_0 \end{cases}$$

Then the average ranking $\bar{r}_{j\cdot} = \sum_{k=1}^{K} r_{j,k}/K$ of $\boldsymbol{x}_j$ across $K$ independent runs forms two distinct clusters, $\mathcal{C}_0$ for $\boldsymbol{X}_{\mathcal{S}_0}$ and $\mathcal{C}_1$ for $\boldsymbol{X}_{\mathcal{S}_1}$. Specifically, $\bar{r}_{j\cdot}$ for $\boldsymbol{X}_{\mathcal{S}_0}$ are expected to cluster in $\mathcal{C}_0$ with mean $(1 + p_0)/2$, while those for $\boldsymbol{X}_{\mathcal{S}_1}$ tend to cluster in $\mathcal{C}_1$ with mean $(p_0 + 1 + p)/2$. Although both cluster means are unknown due to the unknown sparsity $p_0$, their separation can be identified using clustering techniques.

To illustrate, consider the extended Feynman I-38-12 dataset (defined in Section 5.2) with $p = 204$ features, of which

$p_0 = 4$ are relevant. Without loss of generality, we assume that the relevant features $\boldsymbol{X}_{\mathcal{S}_0}$ are $\boldsymbol{x}_1, \boldsymbol{x}_2, \boldsymbol{x}_3, \boldsymbol{x}_4$, i.e., $\mathcal{S}_0 = \{1, 2, 3, 4\}$ and $\mathcal{S}_1 = \{5, \ldots, 204\}$. When $K = 20$ independent BART models are trained on the dataset, the rankings $r_{1,k}, r_{2,k}, r_{3,k}, r_{4,k}$ frequently fall within $\{1, 2, 3, 4\}$ across all $k = 1, \ldots, 20$ runs. This is because the relevant features are frequently selected for tree splits due to their strong associations with the response variable $\boldsymbol{y}$, leading to high VIPs and consistently top rankings. In contrast, irrelevant features $\boldsymbol{x}_5, \ldots, \boldsymbol{x}_{204}$ are included sporadically in BART, with $r_{5,k}, \ldots, r_{204,k}$ distributed randomly across $\{5, \ldots, 204\}$. As evident in Figure 5 in Appendix D.1, the average VIP rankings $\bar{r}_j$ of the relevant features form a low-mean cluster $\mathcal{C}_0$ with a cluster mean of $(1 + p_0)/2 = 2.5$, while those of the irrelevant features form a high-mean cluster $\mathcal{C}_1$, concentrating around $(p_0 + 1 + p)/2 = 104.5$.

However, the sparse regression setting naturally leads to a class imbalance problem as $|\mathcal{C}_0| = p_0$ is much smaller than $|\mathcal{C}_1| = p - p_0$. To this end, we propose to apply agglomerative hierarchical clustering (AHC) with Euclidean distance and average linkage to $(\bar{r}_1, \ldots, \bar{r}_{p\cdot})$ and cut the dendrogram to form two clusters: $\widehat{\mathcal{C}}_0$ and $\widehat{\mathcal{C}}_1$. Then, features in $\widehat{\mathcal{C}}_0$ are retained, while those in $\widehat{\mathcal{C}}_1$ are discarded. Notably, the proposed data-driven selection criterion does not require any knowledge about the sparsity level $p_0$ or a tunable selection threshold. An ablation study evaluating the effect of different clustering algorithms on selection accuracy is available in Appendix D.3. We herein refer to this variable selection method for SR pre-screening as PAN; see Appendix C.2 for implementation details.

## 5. Experiment Design

Using an open-source benchmarking platform, SRBench, we evaluate the PAN+SR framework on two separate tasks. First, we assess its ability to make accurate predictions on "black-box" regression problems in which the underlying regression function remains unknown. Second, we test PAN+SR's ability to find the correct data-generating function $f_0$ on synthetic datasets with known data-generating functions originating from *Feynman Lectures on Physics* (Feynman et al., 2010; Udrescu & Tegmark, 2020).

The experiment settings are summarized in Table 1. All experiments were run on a heterogeneous cluster. Each algorithm was trained on each dataset in 10 repeated trials with a different random state to control both the train/test split and the seed of the algorithm. Each run was performed until a 24-hour time limit was reached or up to 500,000 expression evaluations for black-box problems or 1,000,000 for ground-truth problems. For ground-truth problems, we chose a few representative algorithms in the black-box problems and investigated additional settings of sample size and

signal-to-noise ratio. Datasets were split 75%/25% in training and testing. For black-box problems, hyperparameters were either set to the optimal values published by SRBench or to values recommended by the original authors of the respective methods. The best hyperparameter settings in black-box regression problems were used in ground-truth problems. Instructions for reproducing the experiment is available in Appendix A, and detailed experimental settings are described in Appendix C.

### 5.1. Symbolic Regression Methods

Here we summarize the SR methods evaluated in this paper. A long strand of SR methods is based on genetic programming (GP), a technique for evolving executable data structures, such as expression trees. The most vanilla version we test is gplearn (Stephens, 2020), which performs random expression proposal and iterates through the steps of tournament selection, mutation, and crossover. Advanced GP-based methods utilize different evolutionary strategies and optimization objectives, ranging from Pareto optimization for efficient trade-offs between accuracy and model complexity to program semantics optimization for increasing coherence in expression. Here we test an array of advanced GP-based SR algorithms, including Age-Fitness Pareto optimization (AFP) (Schmidt & Lipson, 2010), AFP with co-evolved fitness estimate (AFP_FE) (Schmidt & Lipson, 2010), Epigenetic Hill Climber (EHC) (La Cava et al., 2014), $\varepsilon$-lexicase selection (EPLEX) (La Cava et al., 2019a), Feature Engineering Automation Tool (FEAT) (La Cava et al., 2019b), Fast Function Extraction (FFX) (McConaghy, 2011), GP version of Gene-pool Optimal Mixing Evolutionary Algorithm (GP-GOMEA) (Virgolin et al., 2021), Interaction-Transformation Evolutionary Algorithm (ITEA) (de Franca & Aldeia, 2021), Multiple Regression Genetic Programming (MRGP) (Arnaldo et al., 2014), Operon (Burlacu et al., 2020), PySR (Cranmer, 2023), and Semantic Back-propagation Genetic Programming (SBP-GP) (Virgolin et al., 2019).

Additional methods include Bayesian Symbolic Regression (BSR) (Jin et al., 2020), which places a prior on the expression tree; Deep Symbolic Regression (DSR) (Petersen et al., 2021), Unified Deep Symbolic Regression (uDSR) (Landajuela et al., 2022), and Dynamic Symbolic Network (DySymNet) (Li et al., 2024) utilize recurrent neural networks to propose symbolic expressions; Transformer-based Planning for Symbolic Regression (TPSR) (Shojaee et al., 2023) leverages pretrained transformer models; AIFeynman 2.0 (Udrescu et al., 2020) which uses a divide-and-conquer technique to recursively decomposing complex problems into lower-dimensional sub-problems.

Table 1: Settings used in the experiments.

| SETTING | BLACK-BOX PROBLEMS | GROUND-TRUTH PROBLEMS |
|---|---|---|
| # OF DATASETS | 35 | 100 |
| # OF ALGORITHMS | 19 | 19 |
| # OF TRIALS PER DATASET | 10 | 10 |
| TRAIN/TEST SPLIT | .75/.25 | .75/.25 |
| TERMINATION CRITERIA | 500K EVALUATIONS OR 24 HOURS | 1M EVALUATIONS OR 24 HOURS |
| SAMPLE SIZE | ALL | 500, 1000, 1500, 2000 |
| SIGNAL-TO-NOISE RATIO | NONE | 0.5, 1, 2, 5, 10, 15, 20, NONE |
| TOTAL COMPARISONS | 12250 | 142000 |
| COMPUTATION COST | 34K CORE HOURS | 104K CORE HOURS |
| MEMORY ALLOCATION | 16 GB | 16 GB |

## 5.2. Datasets

We curated a database of high-dimensional regression problems for testing the capability of PAN+SR. We selected 35 black-box regression problems available in PMLB v1.0 (Romano et al., 2021) using the following criteria: $n < 200$ and $p \geq 10$ or $n \geq 200$ and $p \geq 20$. These problems were used in SRBench and overlap with various open-source repositories, including OpenML (Vanschoren et al., 2014) and the UCI Machine Learning Repository (Kelly et al., 2013).

We also curated 100 high-dimensional ground-truth regression problems by modifying the Feynman Symbolic Regression Database (Udrescu & Tegmark, 2020) to include irrelevant features and white noise. For each equation $f_0(\cdot)$ in the *Feynman Lectures on Physics*, we generated the relevant features $X_{S_0}$ following Udrescu and Tegmark (2020):

$$(x_{1,j}, \ldots, x_{n,j}) \overset{\text{iid}}{\sim} \text{Unif}(a_j, b_j), \quad \text{for } 1 \leq j \leq p_0, \quad (4)$$

where $p_0 = |S_0|$ is the number of relevant features, $n$ is the sample size, and $a_j$ and $b_j$ are the lower and upper bounds for feature $x_j$ described in Udrescu and Tegmark (2020). To study the effect of noise on PAN+SR, we tuned the signal-to-noise ratio (SNR) by adding a Gaussian error term when generating the response variable:

$$y_i = f_0(x_{i,1}, \ldots, x_{i,p_0}) + \varepsilon_i, \quad \text{for } 1 \leq i \leq n, \quad (5)$$

where $\varepsilon_i \overset{\text{iid}}{\sim} N(0, \sigma_\varepsilon^2)$, $\sigma_\varepsilon^2 = \sigma_f^2/\text{SNR}$. When $\sigma_\varepsilon^2 = 0$ or SNR $= \infty$, (4) and (5) generate the original Feynman Symbolic Regression Database.

In addition to the relevant features $X_{S_0} = (x_1, \ldots, x_{p_0})$, we included an array of irrelevant features $X_{\text{irr}}$, representing the era of big data where all reasonable features are included in the dataset. Specifically, for each relevant feature $x_j, j \in S_0$, we generate $(x_{j,\text{irr}}^1, \ldots, x_{j,\text{irr}}^s) \overset{\text{iid}}{\sim} \text{Unif}(a_j, b_j)$, representing $s$ copies of independent and irrelevant features coming from the same distribution as $x_j$. Then, the final feature matrix is $X = [X_{S_0}, X_{\text{irr}}^1, \ldots, X_{\text{irr}}^{p_0}] \in \mathbb{R}^{n \times p}$, where

$X_{\text{irr}}^j = (x_{j,\text{irr}}^1, \ldots, x_{j,\text{irr}}^s) \in \mathbb{R}^{n \times s}$ is the irreverent feature matrix induced by the $j$th relevant feature for $j = 1, \ldots, p_0$, totaling $p = p_0(1+s)$ features. In Section 6.2, we fix $s = 50$ so the total number of features is $p = 51p_0$. Additional dataset information and sampling process are available in Appendix B.

Besides the 3,200 distinct simulation settings described in Table 1 (100 datasets, 8 SNRs, and 4 sample sizes), we include additional simulation settings in Appendix D.4 to further assess PAN+SR's behavior under alternative feature structures. These include (1) additive noise in features, (2) duplicated features, and (3) correlated features.

## 5.3. Metrics

**Predictive Accuracy** We assessed predictive accuracy using the coefficient of determination, defined as

$$R^2 = 1 - \frac{\sum_{i=1}^{n} (y_i - \widehat{y}_i)^2}{\sum_{i=1}^{n} (y_i - \bar{y})^2}.$$

**Model Complexity** In line with SRBench, we define model complexity as the total number of mathematical operators, features, and constants in the model. To avoid redundancy, symbolic models are first simplified using SymPy (Meurer et al., 2017), a Python library for symbolic mathematics.

**Solution Criteria** For ground-truth regression problems, we follow SRBench's definition of symbolic solution. A model $\widehat{f}(X)$ is considered a solution to the SR problem of $y = f_0(X) + \varepsilon$ if $\widehat{f}(X)$ does not reduce to a constant and (1) $\widehat{f} - f_0 = a$ for some $a \in \mathbb{R}$ or (2) $\widehat{f}/f_0 = b$ for some $b \neq 0$. That is, the predicted model $\widehat{f}$ only differs from the true model $f_0$ by either an additive or a multiplicative constant.

While predictive accuracy can be influenced by the simulation design, the symbolic solution criterion offers a more reliable metric for assessing whether an SR method can uncover the true data-generating process. However, since

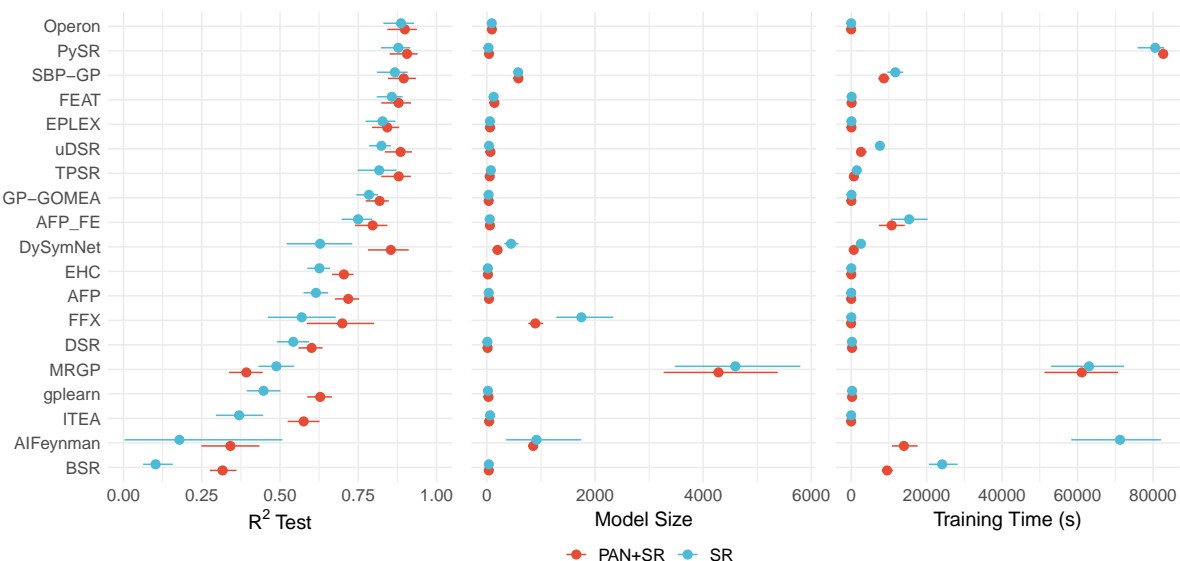

Figure 1: Results on the black-box regression problems. Points indicate the mean test set performance and bars represent the 95% confidence intervals. Training time for PAN+SR includes the runtime of PAN, which averages only 74.14 seconds.

SymPy's simplification process is not always optimal, it is possible that some symbolic solutions are not identified in the process.

**Feature Usage Accuracy** The irrelevant features present a unique challenge for SR methods to identify the correct data-generating model $f_0$. When the predictive model $\widehat{f}$ includes irrelevant features (FPs), it cannot be considered a symbolic solution to $f_0$. Conversely, if $\widehat{f}$ excludes some relevant features (FNs), it also fails to meet the symbolic solution criteria. Although neither FPR nor FNR corresponds directly to symbolic solution rate, they can provide insights into why $\widehat{f}$ does not qualify as a symbolic solution.

## 6. Results

### 6.1. Blackbox Datasets

Figure 1 shows that PAN+SR consistently improves test set $R^2$ across 18 out of 19 SR algorithms, with the largest gains observed in lower-performing methods such as BSR, AIFeynman, and ITEA. For top-performing SR algorithms, the improvements are more modest due to the natural upper limit of $R^2$, but the uplift remains significant. For instance, PAN boosted uDSR from 14th to 5th place in the overall ranking and to 2nd among the standalone SR methods. Furthermore, these $R^2$ improvements are not accompanied by increased model complexity. In some cases, PAN+SR even reduces model complexity, enhancing both parsimony and interpretability.

In addition to accuracy gain, PAN+SR significantly reduces training times for several SR algorithms, including SBP-GP, uDSR, AFP_FE, AIFeynman, and BSR. Notably, AIFeynman, the 2nd slowest running SR algorithm, achieves a 5-fold speedup (from 71250 seconds to 13997 seconds), while uDSR benefits from nearly a 3-fold speedup (from 7628 seconds to 2612 seconds) with PAN pre-screening. The computational overhead introduced by PAN is minimal, averaging only 74.14 seconds on a single core. As PAN relies on independent MCMC chains, this overhead can be further reduced through parallel processing, making PAN+SR both efficient and scalable.

### 6.2. Ground-truth Datasets

Figure 2 summarizes performance on the ground-truth regression problems with $n = 1000$, SNR $= \infty$, and $s = 50$. Methods are sorted by their standalone $R^2$ on the test set. PAN+SR consistently improves both $R^2$ and solution rate across all 19 SR methods. Due to the high dimensionality of the ground-truth problems, the standalone AIFeynman encountered out-of-memory errors and failed to complete any of the 1000 runs. However, PAN significantly improves AIFeynman's performance, lifting it from last place to 2nd overall in symbolic solution rate. Furthermore, PAN consistently outperforms all other nonparametric variable selection methods tested, achieving the highest TPR among four other methods and delivering the best $R^2$ when paired with SR, as detailed in Appendix D.2. This underscores the effectiveness and necessity for nonparametric pre-screening in

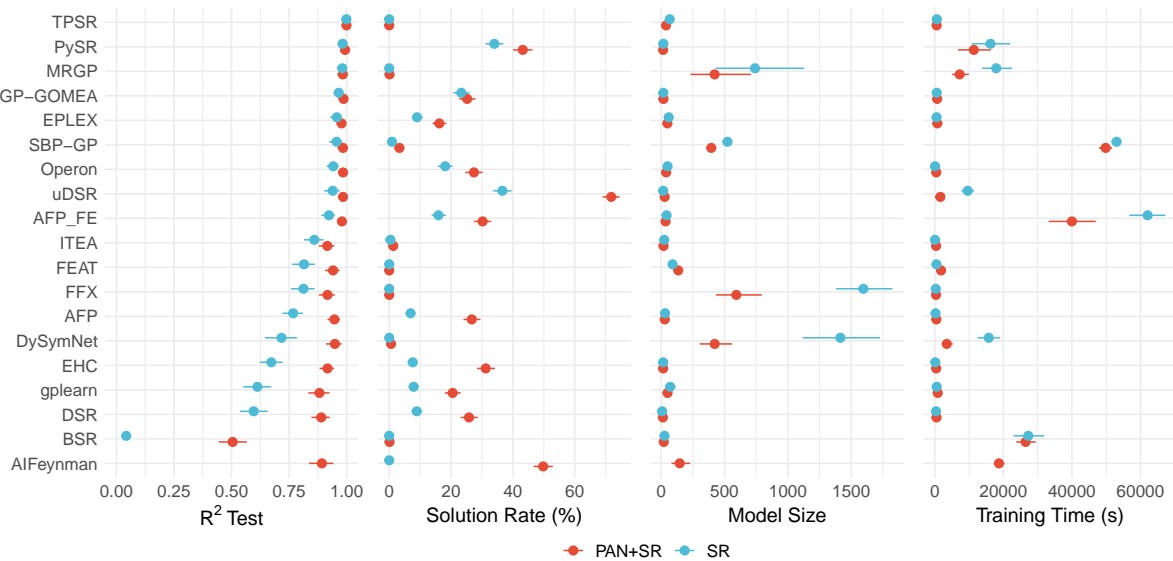

Figure 2: Results on the ground truth regression problems with $n = 1000$, SNR $= \infty$, and $s = 50$. Points indicate the mean test set performance and bars represent the 95% confidence intervals. Training time for PAN+SR includes the runtime of PAN, which averages 325 seconds. AIFeynman fails to complete any run in the standalone setting.

high-dimensional SR problems.

Similar to our findings in the black-box regression problems, this performance gain is not driven by increased model size, and PAN's average computational overhead of 325 seconds remains insignificant to many SR methods. Remarkably, uDSR benefited from nearly a 6-fold speedup with PAN (from 9573 seconds to 1596 seconds) while almost doubling its solution rate (from 36.6% to 71.8%), making it the best performer in solution rate. Additionally, PAN elevated several mid-tier performers such as Operon, AFP_FE, AFP, and EHC, enabling them to surpass the 4th place method, GP-GOMEA, in the standalone SR solution rate ranking.

Beyond the specific simulation setting of $n = 1000$ and SNR $= \infty$, we also investigated the sensitivity of PAN+SR across a range of sample sizes and SNR. In particular, we evaluated PAN+SR with all combinations of sample size $n \in \{500, 1000, 1500, 2000\}$ and SNR $\in \{0.5, 1, 2, 5, 10, 15, 20, \infty\}$. Given the extreme computational burden, we select Operon, the best-performing algorithm in black-box regression problems, to be the SR module for the sensitivity analysis.

Figure 3a demonstrates that both Operon and PAN+Operon maintain consistently lower FPR across all settings of $n$ and SNR, with negligible differences between them. This low FPR reflects the rare inclusion of irrelevant features in the final symbolic models. In noisy settings, we notice a significant increase in PAN's FPR, from 0% at SNR $= \infty$ to over 30% at SNR $= 0.5$. While this noise sensitivity could

be a concern for typical variable selection applications, it is crucial to emphasize that PAN's primary objective is to scale up SR methods by reliably identifying a superset of the relevant features $\mathcal{S}_0$. In this context, minimizing FNs during pre-screening is more critical than avoiding FPs.

Figure 3b illustrates that PAN achieves a near 0% FNR across most simulation settings, highlighting its ability to identify a superset of the true feature set $\mathcal{S}_0$. This is crucial to ensure that the pre-screened dataset $(\boldsymbol{y}, \boldsymbol{X}_{\widehat{\mathcal{S}}})$ used for subsequent SR modeling is comprehensive enough to generate the correct expression $f_0$. However, in the most extreme case, where $n = 500$ and SNR $= 0.5$, PAN's FNR rises to over 5%, and caution is advised when relying on PAN in such cases. On the other hand, the standalone Operon often fails to include all relevant features in its final models across all $n$ and SNR settings, while PAN consistently lower Operon's FNR, enhancing its chance to identify the true function $f_0$. Even with PAN, Operon fails to achieve the best-case FNR set by PAN, particularly under noisy conditions. This elevated FNR negatively impacts Operon's solution rate. For example, changing SNR from $\infty$ to 10, PAN+Operon's average solution rate drops from 27.4% to 0%, and Operon's solution rate falls from 18.1% to 0%. As La Cava et al. (2021) noted, this limitation persists even when Operon is provided with only the relevant features $\boldsymbol{X}_{\mathcal{S}_0}$ and under favorable conditions ($n = 100{,}000$ and SNR $= 100$), indicating that the issue lies beyond PAN pre-screening. Other performance metrics of this sensitivity analysis are available in Appendix D.5.

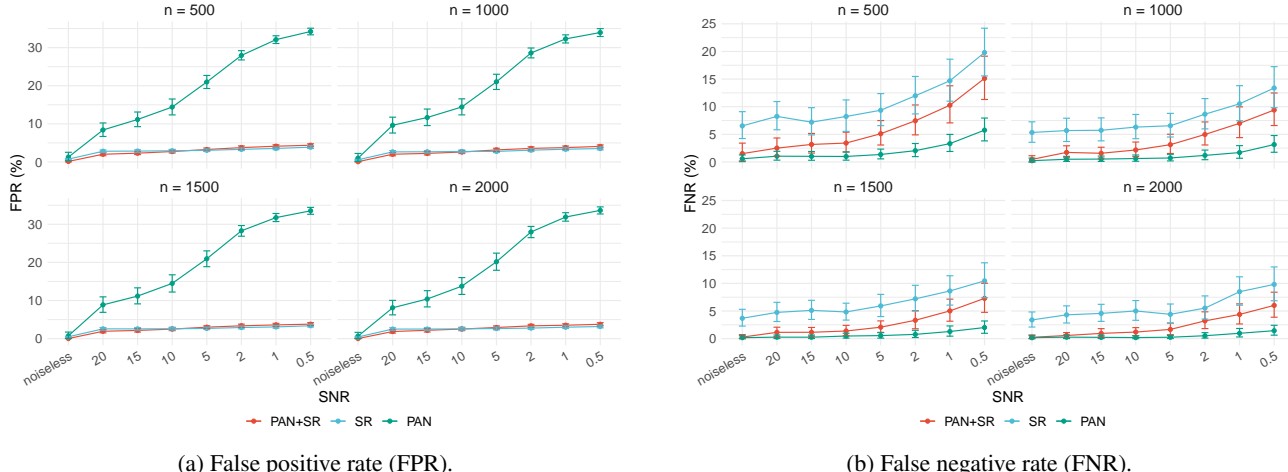

(a) False positive rate (FPR).

(b) False negative rate (FNR).

Figure 3: FPR and FNR of Operon, PAN+Operon, and PAN on the ground truth datasets. PAN refers to the proposed selection method in Section 4. Points indicate the mean performance and bars represent the 95% confidence intervals.

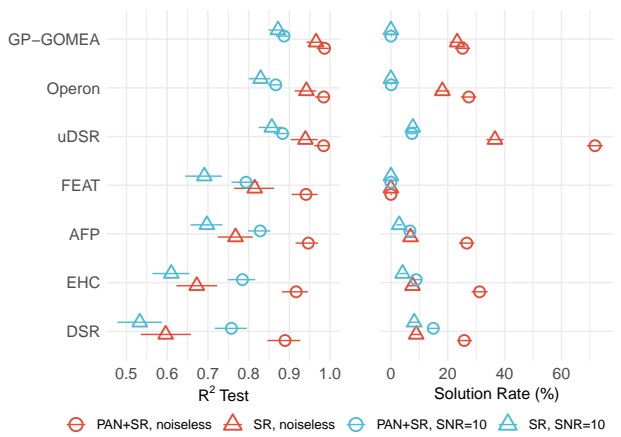

Figure 4: Results of selected methods on the ground truth problems with $n = 1000$, SNR $\in \{\infty, 10\}$, and $s = 50$. Points indicate the mean test set performance and bars represent the 95% confidence intervals.

Beyond Operon, we also evaluated several top-performing SR methods on the ground-truth problems using $n = 1000$ and SNR $\in \{\infty, 10\}$. As shown in Figure 4, PAN+SR consistently improves SR methods across all SNR levels, though all SR and their PAN-boosted variants become less accurate at SNR $= 10$, indicating the challenge when noise is present. In particular, GP-GOMEA performs similarly to Operon, with its solution rate dropping to 0% at SNR $= 10$ for both the standalone and PAN-boosted variants. The best-performing SR algorithm, uDSR, also exhibits vulnerability to noise, with its PAN-boosted solution rate falling from 71.8% to 7.4%. Surprisingly, PAN significantly benefits

DSR, the weakest SR algorithm in Figure 4, increasing its solution rate from 8.2% to 14.9% at SNR $= 10$ and from 8.9% to 25.8% at SNR $= \infty$. These findings highlight the fundamental challenges noise introduces to SR algorithms. To date, SR algorithms have been predominantly developed for noiseless or high-SNR settings, even for "small $p$" problems. We expect that iterative application of the proposed variable selection method, similar to Ye et al. (2024), along with careful consideration of the challenges in extreme-scale SR, could improve performance in low-SNR settings. This will be explored in future work.

## 7. Discussion

In this paper, we introduce PAN+SR, a novel framework designed to address the scalability challenges faced by SR methods when applied to high-dimensional datasets. The growing prevalence of big data necessitates tools capable of efficiently handling such complexity, and PAN+SR addresses this need by integrating a nonparametric pre-screening mechanism with SR. This integration enables the framework to focus the model search on a relevant subset of features, reducing computational burden and improving accuracy.

The core innovation of PAN+SR lies in its nonparametric variable selection method, which filters the input dataset to reduce dimensionality before applying SR. A key challenge in this process is minimizing the risk of false negatives (FNs), where relevant features are mistakenly excluded. Such omissions can critically impair SR methods, as the success of SR depends on having access to the true feature set. To address this issue, we developed a variable selection method designed to ensure that the selected features

form a superset of the true feature set, effectively minimizing the FNR. Our approach leverages the characteristics of VIP rankings derived BART, providing a tuning-free, data-driven variable selection criterion capable of retaining relevant features while excluding irrelevant ones. By preserving a comprehensive set of candidate features, PAN+SR maximizes the likelihood of identifying the true underlying model.

We evaluated PAN+SR across a diverse set of datasets, including 35 high-dimensional real-world datasets from the PMLB database and 100 modified simulated datasets based on the Feynman Symbolic Regression Database. The results were highly promising: PAN+SR improved the performance of 18 out of 19 SR methods on real datasets and all 19 methods on simulated datasets when noise is absent. These findings underscore the framework's potential to enhance the robustness and scalability of SR methods across diverse datasets.

In addition, we explored the sensitivity of PAN+SR to varying sample sizes and SNR. Our analysis demonstrated that the performance gains achieved by PAN+SR are consistent across different sample sizes and remain robust in the presence of noise. Like our extended Feynman database, SDSR (Matsubara et al., 2024) augments the original Feynman database with irrelevant features, bringing the synthetic benchmarks closer to real-world scientific process. However, SDSR adds only 1-3 irrelevant variables, while our setup introduces 100-450 irrelevant variables, posing a substantially more challenging test for both variable selection and symbolic regression. Nonetheless, SDSR rectifies several physical inconsistencies present in the original Feynman benchmark, such as a more realistic treatment of constants and integer-valued variables, and a more careful specification of sampling ranges. Our investigation extends beyond ground-truth datasets by incorporating black-box datasets, thereby mitigating, to some extent, the limitations inherent in purely simulated data. Still, we view SRSD as a valuable and complementary benchmark and plan to incorporate its refinements in future evaluations. In summary, PAN+SR provides a significant step forward in enabling SR methods to handle the complexities of modern datasets, offering improved performance and scalability across a wide range of applications.

## Impact Statement

This paper presents work whose goal is to advance the field of Machine Learning. There are many potential societal consequences of our work, none of which we feel must be specifically highlighted here.

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

## A. Reproducing the Experiment

The experiment made use of an existing symbolic regression (SR) benchmarking platform, SRBench (La Cava et al., 2021), and changes were made to facilitate other functionalities, including signal-to-noise ratio (SNR) tuning, feature pre-screening, and variable usage accuracy calculation. The README file in our GitHub repository https://github.com/mattsheng/PAN_SR details the complete set of commands for reproducing the experiment. Here, we provide a short summary of the experiment process. Experiments are launched from the `experiments/` folder via the script `analyze.py`. After installing and configuring the conda environment provided by SRBench, the complete black-box experiment on standalone SR methods can be started via the following command:

```
python analyze.py /path/to/pmlb/ \
    -results ../results_blackbox/SR/ \
    -n_trials 10 \
    -time_limit 24:00 \
    -tuned -skip_tuning
```

To enable PAN pre-screening, the users can either specify the path to a pre-run variable selection result or run the pre-screening in place. The first option is useful when the users need to compare different SR methods on the same dataset:

```
python analyze.py /path/to/pmlb \
    -results ../results_blackbox/SR_BART_VIP \
    -n_trials 10 \
    -time_limit 24:00 \
    -vs_method BART_VIP \
    -vs_result_path ../results_blackbox/pmlb_BART_VIP_withidx.feather \
    -vs_idx_label idx_hclst \
    -tuned -skip_tuning
```

If no path is given to `-vs_result_path`, the PAN pre-screening will be run in place. Similarly, the ground-truth experiment for the standalone SR methods on Feynman datasets with a sample size of $n = 1000$ and an SNR of 10 can be run by the following command:

```
python analyze.py /path/to/feynman \
    -results ../results_feynman/SR \
    -signal_to_noise 10 \
    -n 1000 \
    -sym_data  \
    -n_trials 10 \
    -time_limit 24:00 \
    -tuned -skip_tuning
```

Note that `-sym_data` enables more performance metric calculations only available for ground-truth problems. To run PAN pre-screening only on the Feynman datasets with a sample size of $n = 1000$ and an SNR of 10, we can use the following command:

```
python analyze.py /path/to/feynman \
    -script BART_selection \
    -ml BART_VIP \
    -results ../results_feynman/BART_VIP/n_1000/ \
    -signal_to_noise 10 \
    -n 1000 \
    -sym_data \
    -n_trials 10 \
    -rep 20 \
    -time_limit 24:00
```

The `-rep 20` argument instructs the program to run $K = 20$ replications of BART for estimating the variable ranking $r_{jk}$ of the $j$th feature at the $k$th run. Users can use other variable selection methods by modifying the `BART_selection.py` script.

## B. Additional Dataset Information

**PMLB datasets** Black-box datasets and their metadata are available from PMLB under an MIT license and is described in detail in Romano et al. (2021). In this experiment, we only focus on high-dimensional regression datasets available from PMLB. Specifically, we use PMLB regression datasets satisfying the following criteria:

1. $n < 200$ and $p \geq 10$, or

2. $n \geq 200$ and $p \geq 20$.

Furthermore, datasets that have categorical features (number of unique value $\leq 5$) or non-continuous response variable (proportion of unique value $< 0.9$) are excluded since they are incorrectly classified as regression task (Dick, 2022). Among the datasets meeting these criteria, we found that two datasets, 195_auto_price and 207_autoPrice, are identical, and we only kept 195_auto_price in our analysis. See Dick (2022) for a detailed analysis of the dataset duplication and incorrect problem classification issues of PMLB.

**Feynman datasets** The original Feynman database described in Udrescu and Tegmark (2020) consists of only the relevant features $\boldsymbol{X}_{\mathcal{S}_0}$ and a large sample size of $n = 10^5$, and is available in Feynman Symbolic Regression Database (https://space.mit.edu/home/tegmark/aifeynman.html). We extended the Feynman Symbolic Regression Database to include irrelevant features $\boldsymbol{X}_{\text{irr}}^j \in \mathbb{R}^{n \times s}$ for each relevant feature $\boldsymbol{x}_j$, $j \in \mathcal{S}_0$. To take advantage of the SRBench platform, we standardized the Feynman equations to PMLB format and included metadata detailing the true model and the units of each variable. The extended Feynman datasets are generated using the Python script provided in feynman_dataset_code/generate_feynman_dataset.py. To avoid the need to generate different datasets for each sample size $n$ considered in the main paper, we set $s = 50$ and $n = 100,000$ for all Feynman equations with random state control; we refer to this as the full Feynman datasets. In the experiment, the full Feynman datasets are randomly split into a 75%/25% train/test set. If the train set contains more samples than the desired training sample size $n$, the train and test sets will be further subsampled so that $\boldsymbol{X}_{\text{train}}$ has exactly $n$ samples and $\boldsymbol{X}_{\text{test}}$ has exactly $\lfloor n/3 \rfloor$ samples.

Users can also generate datasets using other data-generating functions $f_0$ by supplying a CSV file with the expression of $f_0(\cdot)$ and an additional CSV file describing the desired uniform distribution (i.e., the lower and upper bounds of the distribution) of each variable in $f_0(\cdot)$. See feynman_dataset_code/FeynmanEquations.csv and feynman_dataset_code/units.csv for more details.

**Sampling Process for Extended Feynman Datasets** The sampling process for the extended Feynman datasets is described in the main text and is reproduced here for completeness of the data description in this section.

For each equation $f_0(\cdot)$ in the *Feynman Lectures on Physics*, we generated the relevant features $\boldsymbol{X}_{\mathcal{S}_0}$ following Udrescu and Tegmark (2020):

$$(x_{1,j}, \ldots, x_{n,j}) \overset{\text{iid}}{\sim} \text{Unif}(a_j, b_j), \quad \text{for } 1 \leq j \leq p_0, \tag{6}$$

where $p_0 = |\mathcal{S}_0|$ is the number of relevant features, $n$ is the sample size, and $a_j$ and $b_j$ are the lower and upper bounds for feature $\boldsymbol{x}_j$ described in https://space.mit.edu/home/tegmark/aifeynman/FeynmanEquations.csv. Then, the response variable is generated as follow:

$$y_i = f_0(x_{i,1}, \ldots, x_{i,p_0}) + \varepsilon_i, \quad \text{for } 1 \leq i \leq n, \tag{7}$$

where $\varepsilon_i \overset{\text{iid}}{\sim} N(0, \sigma_\varepsilon^2)$ is an additive Gaussian error, $\sigma_f^2$ denotes the sample variance of $f_0(\cdot)$, and $\sigma_\varepsilon^2 = \sigma_f^2/\text{SNR}$ is the error variance tuned to a prescribed signal-to-noise ratio (SNR). When $\sigma_\varepsilon^2 = 0$ (i.e., SNR $= \infty$), (6) and (7) generate the original Feynman Symbolic Regression Database.

For each relevant feature $\boldsymbol{x}_j$, $j = 1, \ldots, p_0$, we generate $s = 50$ copies of irrelevant features following the distribution of $\boldsymbol{x}_j$: $(\boldsymbol{x}_{j,\text{irr}}^1, \ldots, \boldsymbol{x}_{j,\text{irr}}^s) \overset{\text{iid}}{\sim} \text{Unif}(a_j, b_j)$. Then, the final feature matrix is $\boldsymbol{X} = [\boldsymbol{X}_{\mathcal{S}_0}, \boldsymbol{X}_{\text{irr}}^1, \ldots, \boldsymbol{X}_{\text{irr}}^{p_0}] \in \mathbb{R}^{n \times p}$, where $\boldsymbol{X}_{\text{irr}}^j = (\boldsymbol{x}_{j,\text{irr}}^1, \ldots, \boldsymbol{x}_{j,\text{irr}}^s) \in \mathbb{R}^{n \times s}$ is the irreverent feature matrix induced by the $j$th relevant feature for $j = 1, \ldots, p_0$, totaling $p = p_0(1 + s)$ number of features.

In Appendix D.4, we consider sampling processes where features are not iid sampled from a uniform distribution.

## C. Additional Experiment Details

### C.1. General Experiment Settings

Experiments were run in a heterogeneous cluster composed of nodes with Intel(R) Xeon(R) CPU E5-2620 v2 @ 2.60GHz, Intel(R) Xeon(R) CPU E5-2650 v4 @ 2.20GHz, Intel(R) Xeon(R) Gold 6126 CPU @ 2.60GHz, Intel(R) Xeon(R) Gold 6230 CPU @ 2.10GHz, and AMD EPYC 7642 CPU @ 2.3GHz processors. The training of a single method on a single dataset for a fixed random seed was considered a job. Each job was managed by SLURM Workload Manager to receive one CPU core, 16GB of RAM, and a time limit of 24 hours. For the ground-truth problems, each final model was given an additional 5 minutes for each of the following steps: 1) cleaning the model for SymPy parsing, 2) simplifying the cleaned model using SymPy, 3) checking the difference solution criterion of the simplified model, 4) checking the ratio solution criterion of the simplified model, and 5) calculating model size (complexity). When the simplification of the cleaned model exceeded the 5-minute wall clock, steps 3-5 were run on the cleaned model instead.

### C.2. Implementation Details of the Proposed Variable Selection Method

The proposed method uses the `bartMachine` R package for its BART implementation. For each dataset, we fit $K = 20$ independent BART models and record the ranking $r_{j,k}$ of variable $x_j$'s variable inclusion proportion (VIP) in the $k$th run; the hyperparameters for `bartMachine` are summarized in Table 2. To cluster the VIP rankings into 2 clusters, we use the `hclust` function in R to perform agglomeration clustering (unweighted pair group method with arithmetic mean) on the Euclidean dissimilarity matrix of the VIP rankings. Then, $x_j$ is selected if $\bar{r}_{j.} = \sum_{k=1}^{K} r_{j,k}/K$ belongs to the low-mean cluster.

Table 2: Hyperparameters in `bartMachine`.

| Parameter | Value |
|---|---|
| # of trees | 20 |
| # of burn-in samples | 10,000 |
| # of posterior samples | 10,000 |

## D. Additional Results

### D.1. Visualization of Average VIP Rankings $\bar{r}_{j.}$

Figure 5 shows the average BART VIP rankings for Feynman equation I-38-12 with $n = 1000$. At high SNR, there is a clear separation between the low- and high-mean clusters, and the hypothesized cluster means closely match their actual values. As SNR decreases, irrelevant features tend to receive higher rankings, slightly shifting the cluster means incurring more false positives (FPs). Despite this deviation, the cluster means remain far apart, ensuring separation between relevant and irrelevant features.

Figure 6 further demonstrates the clustering accuracy of the proposed method. Regardless of the SNR level, all true features are consistently assigned to the low-mean cluster, which is highly desirable in the PAN+SR framework. While decreasing SNR leads to some misclassification of the irrelevant features, the proposed method ensures that no true features are excluded. This robustness in retaining the true features under varying noise levels makes the proposed method well-suited for PAN+SR framework and high-dimensional SR tasks.

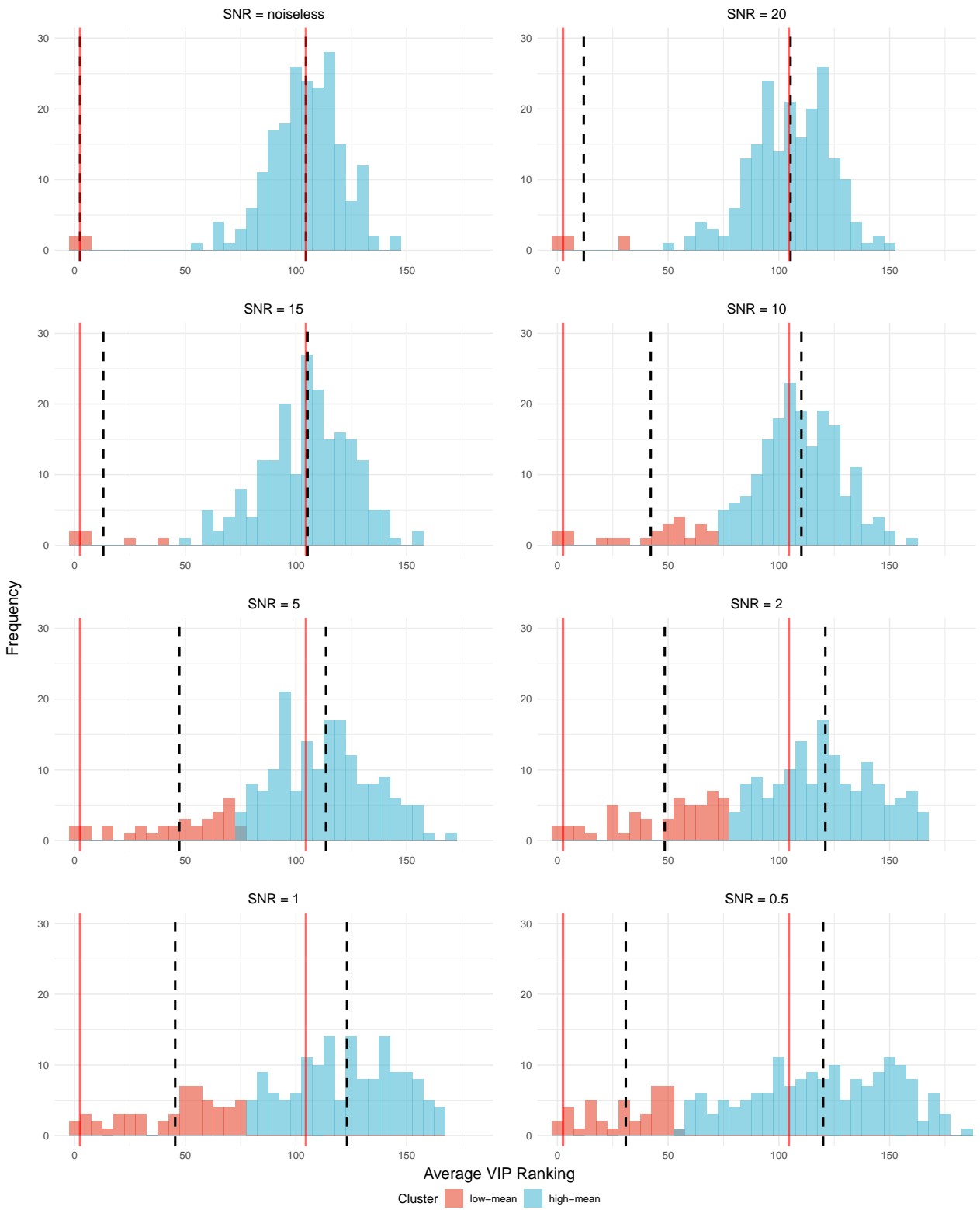

Figure 5: Average BART VIP rankings $\bar{r}_{j\cdot}$ over $K = 20$ runs on Feynman equation I-38-12 with $n = 1000$, $p_0 = 4$, and $p = 204$. Black vertical dashed lines indicate the cluster means. Red solid vertical lines are the hypothesized cluster means: $(1 + p_0)/2 = 2.5$ and $(p_0 + 1 + p)/2 = 104.5$.

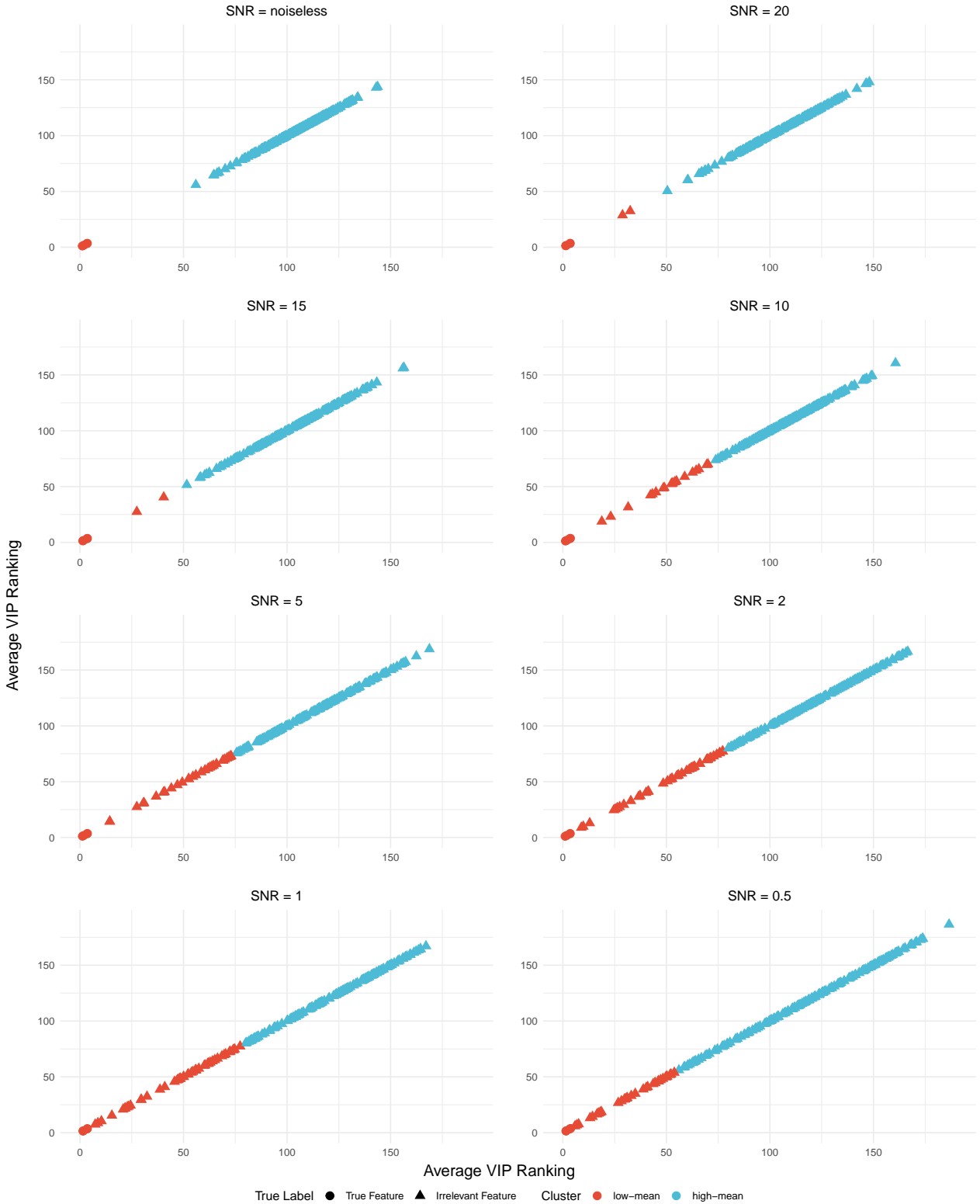

Figure 6: Hierarchical clustering accuracy on Feynman equation I-38-12 with $n = 1000$, $p_0 = 4$, and $p = 204$. Red and teal represent the low- and high-mean clusters, respectively. Circles and triangles represent relevant and irrelevant features, respectively.

## D.2. Analysis of Different Nonparametric Variable Selection Methods

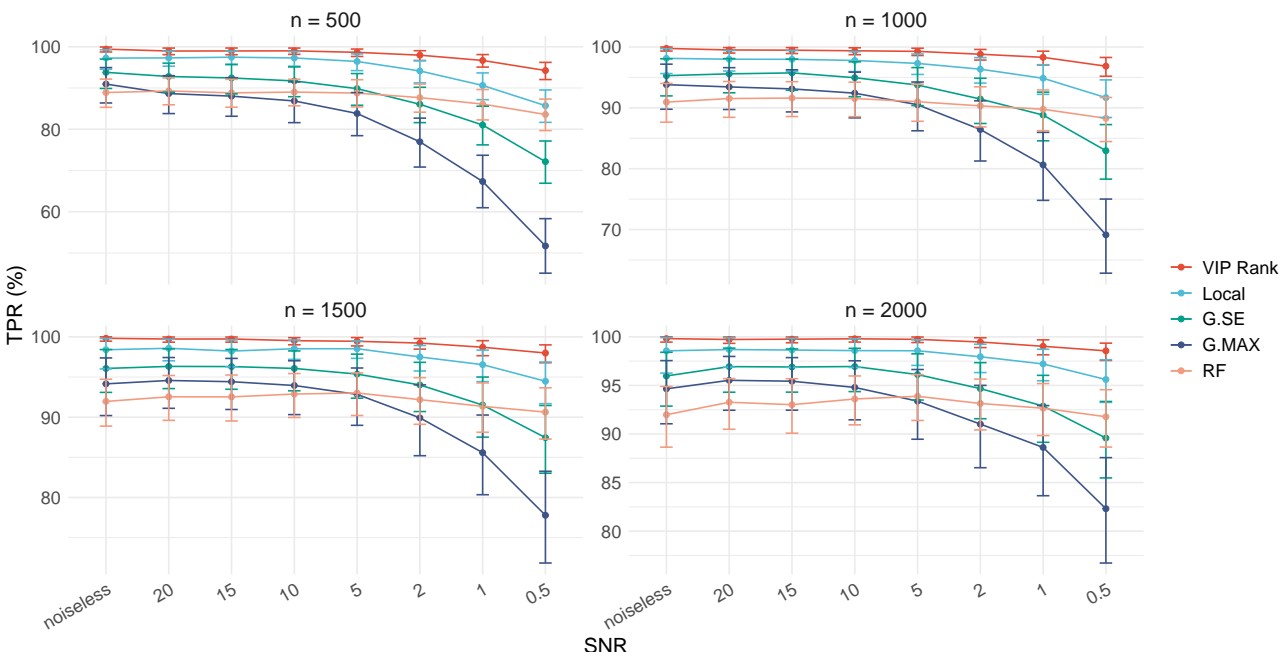

Figure 7: True positive rate (TPR) on the Feynman datasets for $n = 500, 1000, 1500, 2000$ and SNR $= \infty, 20, 15, 10, 5, 2, 1, 0.5$. Points indicate the mean performance, and bars show the 95% confidence interval. VIP Rank is the proposed method for PAN pre-screening. Local, G.SE, G.MAX, and RF are alternative nonparametric variable selection methods.

PAN pre-screening presents a unique challenge to nonparametric variable selection methods where any missed signals (false negative) will eliminate the correct expression $f_0(\cdot)$ from the search space. That is, a true positive rate (TPR) near 100% in the pre-screening phase is necessary to ensure successful SR tasks. Figure 7 compares the average TPR of five nonparametric variable selection methods across various configurations of $n$ and SNR on the Feynman datasets. VIP Rank, the proposed method, is compared with three BART permutation test-based methods (Local, G.SE, and G.MAX) (Bleich et al., 2014) and the Random Forest (RF) variable selection method in PySR (Cranmer, 2023). Of the three BART permutation test-based methods, BART-Local applies the least stringent selection criteria, while BART-G.MAX is the most stringent, with BART-G.SE offering a balance between the two. The RF implementation requires users to specify the number of selected variables $k$, which we tuned over $\{1, 2, \ldots, 20\}$ using 5-fold cross-validation.

VIP Rank consistently achieves the highest TPR, nearing or reaching 100% across all experimental settings. In noiseless conditions (SNR $= \infty$), only VIP Rank attains a perfect TPR of 100%. Although there is a slight TPR decline for VIP Rank at $n = 500$ and SNR $\leq 5$, it still outperforms the other methods, particularly at $n = 500$ and SNR $= 0.5$. These results reinforce the need for a specialized variable selection method for PAN pre-screening. In addition to the four methods considered here, we point readers to Ye et al. (2024), where they analyzed three additional nonparametric variable selection methods and showed that none outperform BART-G.SE in terms of TPR.

Figure 8 illustrates the false positive rate (FPR), a crucial metric for evaluating variable selection accuracy. As discussed in the main paper, VIP Rank produces higher FPR under low SNR conditions–a tradeoff made to maintain a near-perfect true positive rate (TPR). While this tradeoff may be undesirable for typical variable selection tasks, it is acceptable for PAN pre-screening, where minimizing false negatives (FNs) is the priority. The three BART permutation-based methods and RF consistently maintain low and robust FPRs across all settings of $n$ and SNR. However, as Figure 7 shows, this strict control of FPR comes at the cost of worse TPR performance.

To further evaluate the impact of variable selection methods in the PAN+SR framework, we replaced VIP Rank with BART-G.SE and compared their performance using Operon as the SR method. Operon was chosen for this analysis due to

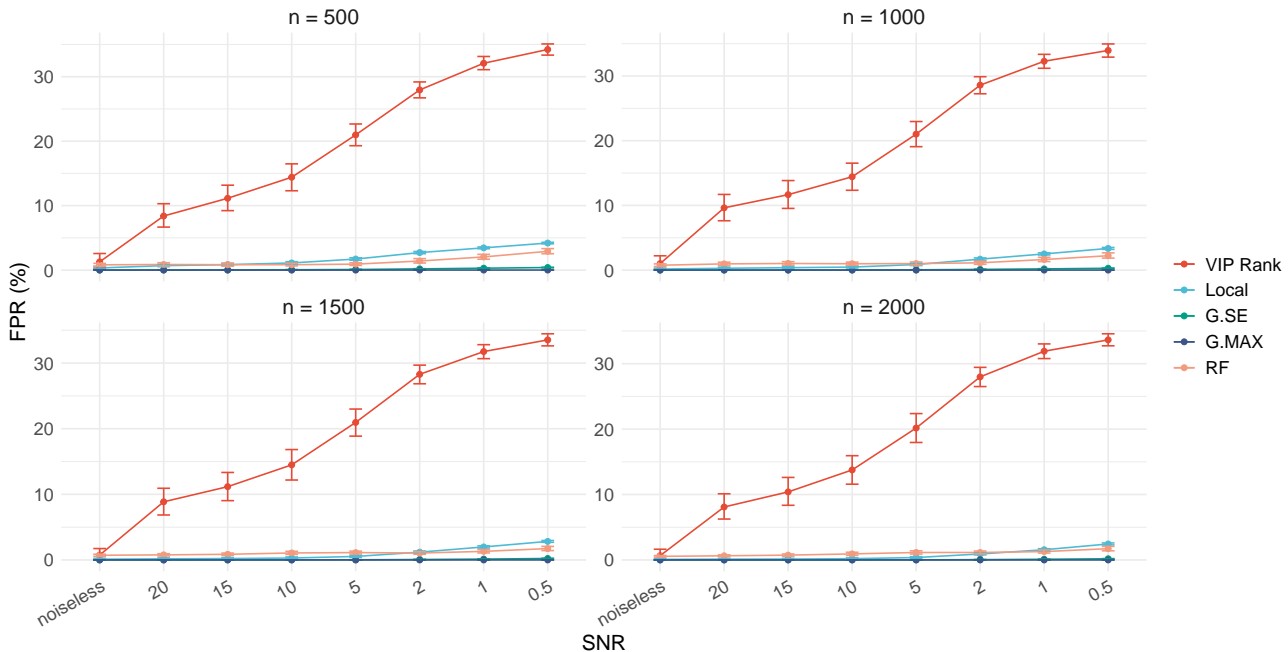

Figure 8: False positive rate (FPR) on the Feynman datasets for $n = 500, 1000, 1500, 2000$ and SNR $= \infty, 20, 15, 10, 5, 2, 1, 0.5$. Points indicate the mean performance, and bars show the 95% confidence interval. VIP Rank is the proposed method for PAN pre-screening. Local, G.SE, G.MAX, and RF are alternative nonparametric variable selection methods.

its strong $R^2$ performance in both the black-box and ground-truth experiments. Table 3 summarizes the average test set $R^2$ on the Feynman dataset. VIP+SR consistently achieves the highest $R^2$ across all experimental settings. For instance, at $n = 500$ and SNR= 20, VIP+SR achieves an average $R^2$ of 0.892, compared to 0.860 for GSE+SR and 0.846 for standalone SR. Under high noise conditions, VIP+SR continues to demonstrate better robustness than GSE+SR. At $n = 500$ and SNR= 0.5, VIP+SR scores 0.145, slightly outperforming GSE+SR (0.142) and standalone (0.142). This trend is consistent across different different sample sizes $n$.

### D.3. Effect of Different Clustering Algorithms

The proposed VIP Rank variable selection method can be implemented using various off-the-shelf clustering algorithms. However, due to the class imbalance nature of the variable selection problem, not all clustering algorithms are suitable. In this ablation study, we examine the effect of clustering algorithms on TPR and FPR performances of VIP Rank. We elected 10 clustering algorithms available in `scikit-learn v1.5.7`: agglomerative hierarchical clustering (AHC), k-means++, Gaussian mixture model (GMM), Birch, Mean Shift, Affinity Propagation, Spectral, OPTICS, HDBSCAN, and DBSCAN.

As illustrated in Figure 9, the first 5 clustering algorithms (AHC, k-mean++, GMM, Birch, Mean Shift) achieve the highest TPR across all simulation settings with indistinguishable differences. Affinity Propagation also has similar TPR compared with the top 5 algorithms but lacks behind in noisy (e.g., SNR $= 0.5$) and small-$n$ (e.g., $n = 500$) settings. The rest of the pack has significantly worse TPR and are thus not suitable in VIP Rank.

Since the top 5 algorithms have indistinguishable TPR, we elect one with the least FPR. As shown in Figure 10, AHC has significantly lower FPR than the rest of the top 5 algorithms across most simulation settings. Combine with its near 100% TPR, AHC is capable of identifying a more compact feature set that has a high probability of containing all relevant features.

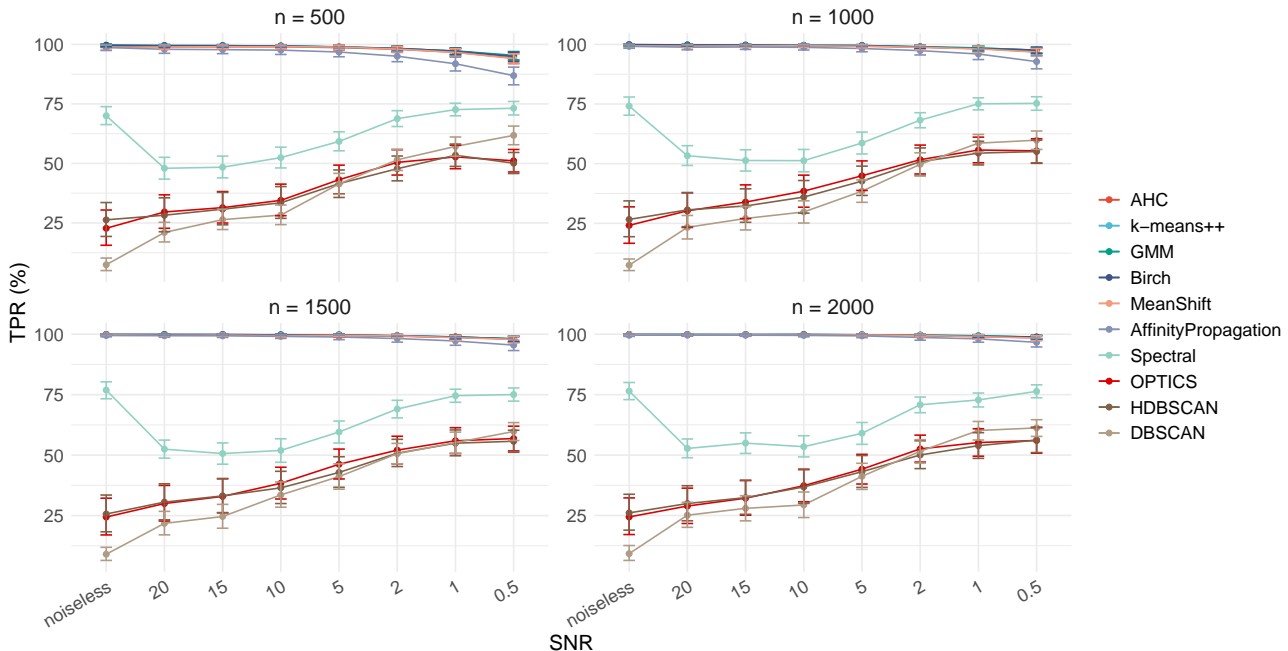

Figure 9: True positive rate of various ablations of clustering algorithm.

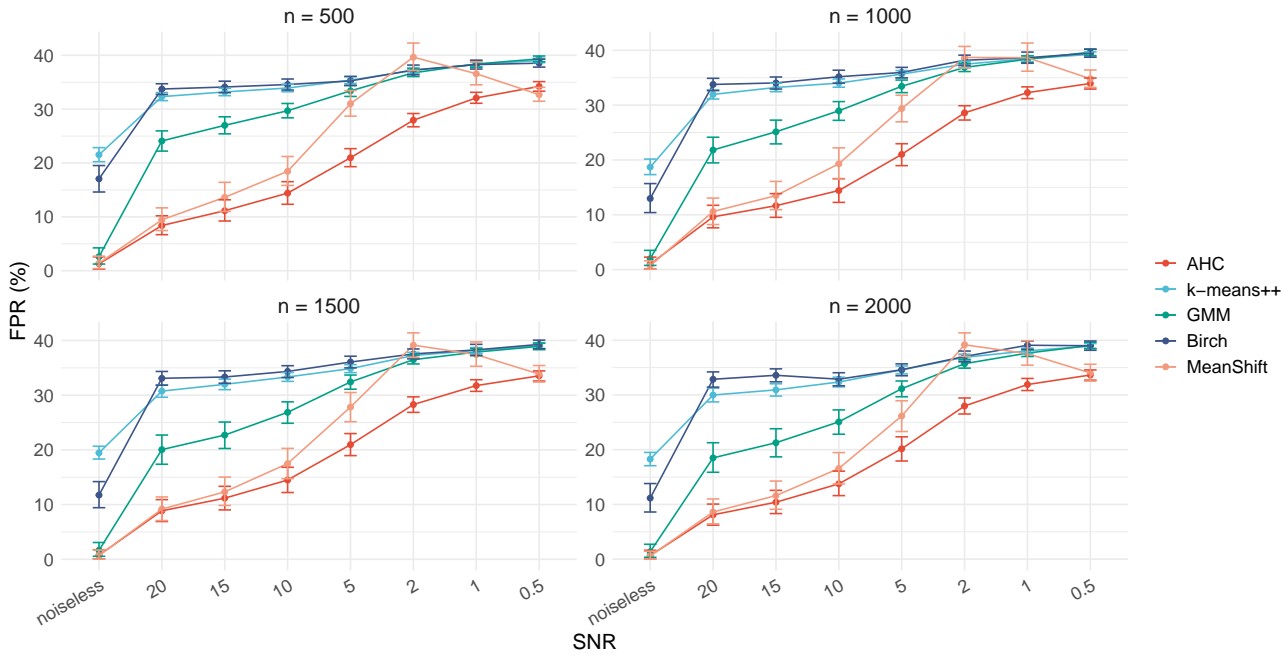

Figure 10: False positive rate of various ablations of clustering algorithm.

Table 3: Average test set $R^2$. The highest value in each experimental setting is in bold.

|  | noiseless | 20 | 15 | 10 | 5 | 2 | 1 | 0.5 |
|---|---|---|---|---|---|---|---|---|
| $n = 500$ | | | | | | | | |
| VIP+SR | **0.974** | **0.892** | **0.870** | **0.837** | **0.730** | **0.525** | **0.335** | **0.145** |
| GSE+SR | 0.948 | 0.860 | 0.859 | 0.818 | 0.710 | 0.510 | 0.327 | 0.142 |
| SR | 0.915 | 0.846 | 0.840 | 0.792 | 0.702 | 0.506 | 0.322 | 0.142 |
| $n = 1000$ | | | | | | | | |
| VIP+SR | **0.984** | **0.919** | **0.901** | **0.867** | **0.774** | **0.586** | **0.406** | **0.229** |
| GSE+SR | 0.971 | 0.914 | 0.897 | 0.851 | 0.774 | 0.574 | 0.405 | 0.229 |
| SR | 0.942 | 0.883 | 0.867 | 0.825 | 0.747 | 0.580 | 0.393 | 0.227 |
| $n = 1500$ | | | | | | | | |
| VIP+SR | **0.990** | **0.928** | **0.909** | **0.874** | **0.792** | **0.612** | **0.433** | **0.260** |
| GSE+SR | 0.961 | 0.910 | 0.899 | 0.866 | 0.781 | 0.600 | 0.428 | 0.257 |
| SR | 0.956 | 0.895 | 0.878 | 0.856 | 0.761 | 0.592 | 0.426 | 0.255 |
| $n = 2000$ | | | | | | | | |
| VIP+SR | **0.990** | **0.935** | **0.914** | **0.887** | **0.805** | **0.619** | **0.448** | **0.277** |
| GSE+SR | 0.963 | 0.918 | 0.905 | 0.872 | 0.787 | 0.617 | 0.445 | 0.272 |
| SR | 0.960 | 0.907 | 0.892 | 0.855 | 0.781 | 0.611 | 0.437 | 0.272 |

## D.4. Effect of Noisy, Duplicated, and Correlated Predictors

In addition to the extensive simulation settings described in Section 5.2, we further evaluate VIP Rank under alternative predictor structures that challenge common modeling assumptions:

- **Baseline:** $x_1, \ldots, x_p \overset{\text{iid}}{\sim} \text{Unif}(0, 1)$

- **Noisy $X$:** Independent Gaussian noise is added to each predictor with variance equal to 1/5 of the signal variance

- **Duplicated $X$:** A redundant feature is added: $x_6 = x_1 + x_2$, where $x_1$ and $x_2$ are relevant predictors

- **Correlated $X$:** $x_1, \ldots, x_p \sim \text{Unif}(0, 1)$ with an autocorrelation structure: $\rho_{ij} = 0.9^{|i-j|}$.

The response variable $y$ is generated according to the Friedman equation (1991):

$$y = 10\sin(\pi x_1 x_2) + 20(x_3 - 0.5)^2 + 10x_4 + 5x_5 + \varepsilon, \qquad \varepsilon \sim N(0, \sigma^2).$$

We fix $n = 1000$, $p = 100$, SNR $= 10$, and repeat each scenarios for 100 trials. Table 4 reports the average TPR and FPR. VIP Rank consistently identifies all relevant features across all scenarios, demonstrating strong robustness to noise, redundancy, and correlation among predictors.

Table 4: Average performance in each scenario across 100 trials.

| Scenario | TPR | FPR |
|---|---|---|
| Baseline | 100% | 10.58% |
| Noisy $X$ | 100% | 26.42% |
| Duplicated $X$ | 100% | 11.11% |
| Correlated $X$ | 100% | 15.98% |

### D.5. Additional Performance Metrics for Operon vs PAN+Operon

Figures 11, 12, 13 show additional metrics not discussed in the main paper. Although PAN+Operon's solution rate plummeted from ~27% at SNR = ∞ to 0% at SNR = 20 across all $n$, Figure 11 shows there is still improvement in $R^2$ on test set across all $n$ and SNR, while improving model interpretability evidenced by the uniformly lower model size in Figure 12.

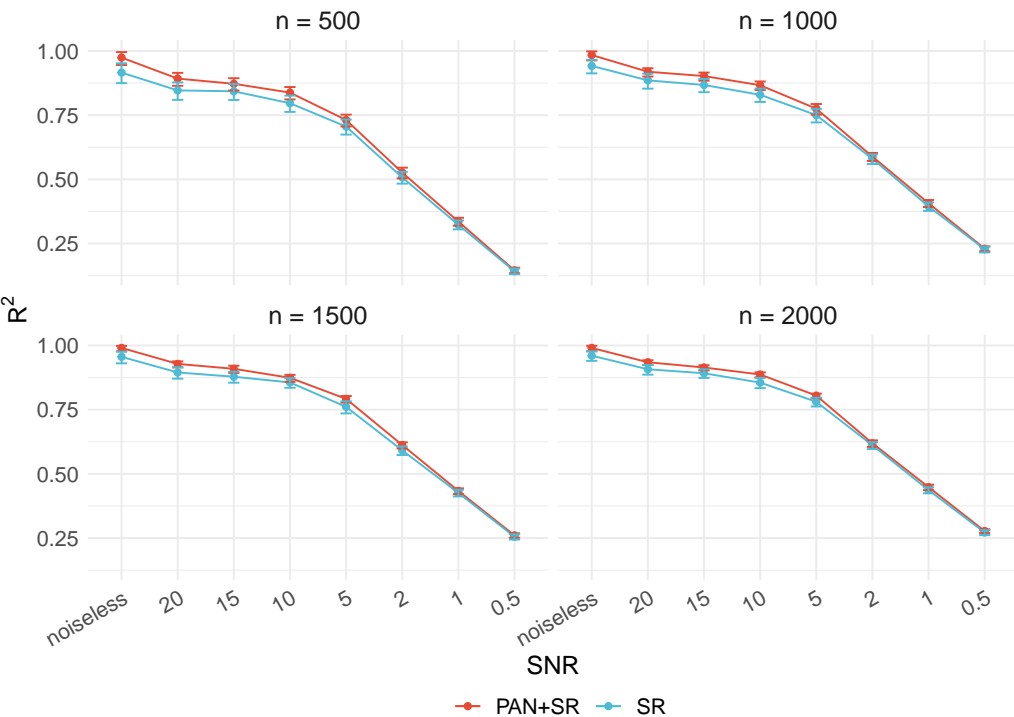

Figure 11: $R^2$ on test set with Operon as the SR module. Points indicate the average $R^2$ on test set and bars represent the 95% confidence intervals.

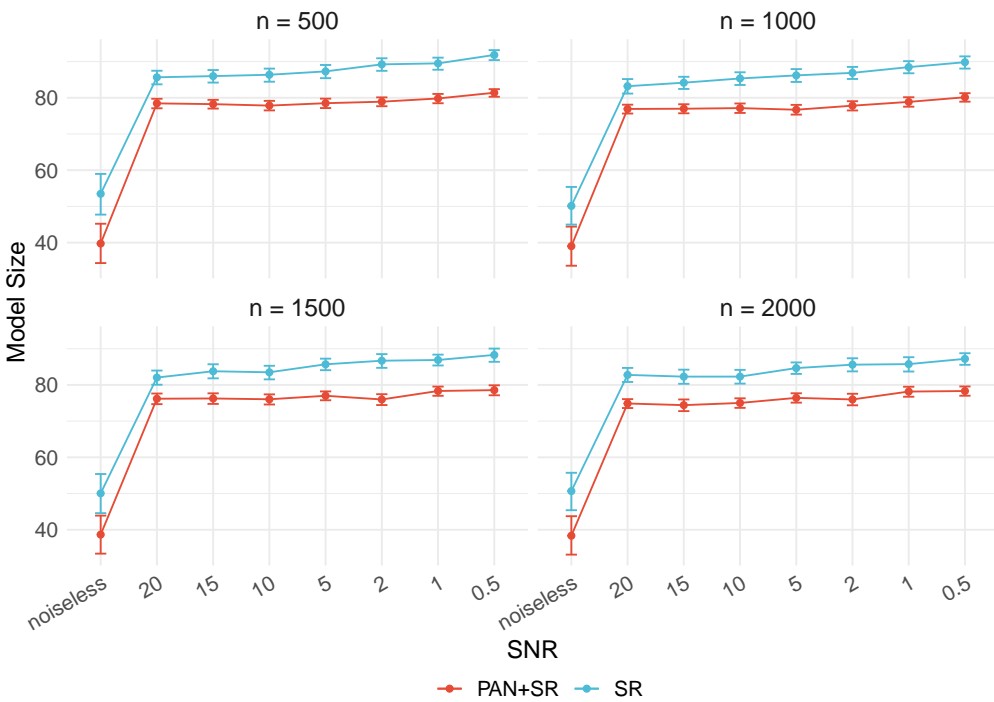

Figure 12: Model size with Operon as the SR module. Points indicate the average model size and bars represent the 95% confidence intervals.

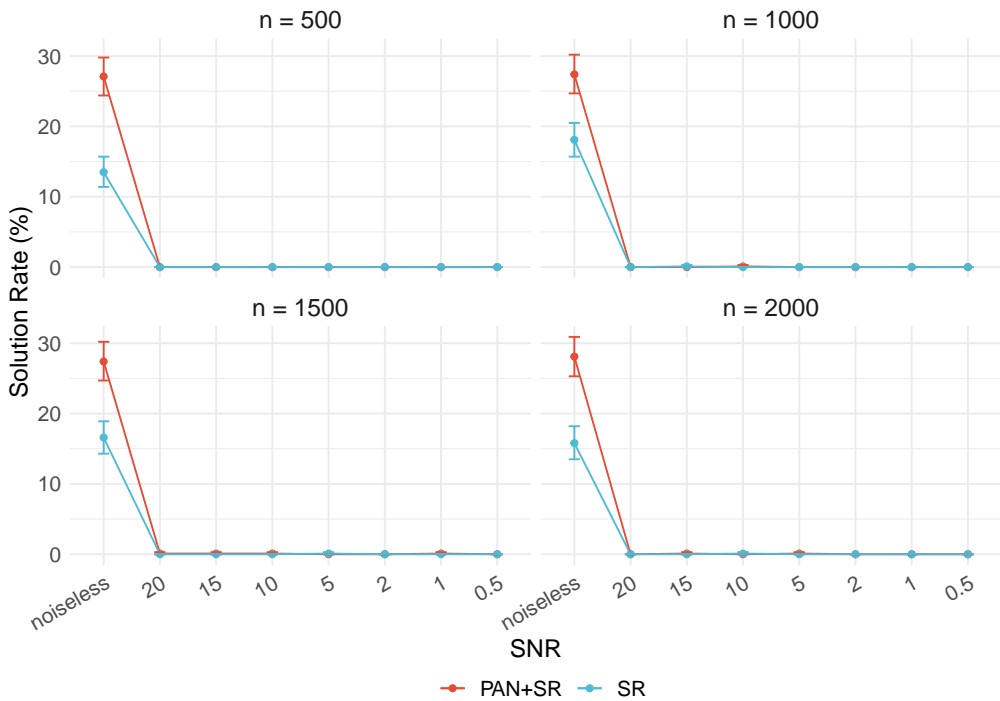

Figure 13: Solution rate with Operon as the SR module. Points indicate the average solution rate and bars represent the 95% confidence intervals.

