# OpenReview forum: "Ab Initio Nonparametric Variable Selection for Scalable Symbolic Regression with Large $p$"
_ICML.cc/2025/Conference — ICML 2025 poster_

### Official Review · Reviewer_pCrp · 2025-03-07

**Overall Recommendation:** 3

**Summary:**

This paper proposes a variable selection method for input variables related to the output. The proposed method is used for data preprocessing in symbolic regression and can improve the accuracy and speed of symbolic regression. Specifically, the authors propose a method called PAN+SR, which combines a key idea of ab initio nonparametric variable selection with SR to efficiently pre-screen large input spaces and reduce search complexity while maintaining accuracy. In the paper, the authors emphasize the importance of FNR, as any erroneous exclusion of related variables will lead to the failure of subsequent symbolic regression. The authors conducted numerous experiments in the paper to demonstrate the effectiveness of the proposed method, including the improvement of regression accuracy and the robustness of the method to noise.

**Claims And Evidence:**

The supporting evidence is insufficient, as the data used in the paper is self constructed and lacks comparison with similar methods.

**Essential References Not Discussed:**

The contribution of the paper lies in proposing a variable selection algorithm rather than a symbolic regression algorithm, but the author only discussed the symbolic regression algorithm in Related Works and did not discuss other variable selection algorithms. Moreover, there was no comparison with other variable selection algorithms in the experiment, which is puzzling. Here are three relevant literature for the author's reference.

1.Q. Chen, M. Zhang and B. Xue, "Feature Selection to Improve Generalization of Genetic Programming for High-Dimensional Symbolic Regression," in IEEE Transactions on Evolutionary Computation, vol. 21, no. 5, pp. 792-806, Oct. 2017, doi: 10.1109/TEVC.2017.2683489.

2.Q. Chen, B. Xue, B. Niu and M. Zhang, "Improving generalisation of genetic programming for high-dimensional symbolic regression with feature selection," 2016 IEEE Congress on Evolutionary Computation (CEC), Vancouver, BC, Canada, 2016, pp. 3793-3800, doi: 10.1109/CEC.2016.7744270.

3.B. Al-Helali, Q. Chen, B. Xue and M. Zhang, "Genetic Programming for Feature Selection Based on Feature Removal Impact in High-Dimensional Symbolic Regression," in IEEE Transactions on Emerging Topics in Computational Intelligence, vol. 8, no. 3, pp. 2269-2282, June 2024, doi: 10.1109/TETCI.2024.3369407.

**Experimental Designs Or Analyses:**

There are some issues with the experimental design of the paper. Firstly, the contribution of the paper lies in proposing a variable selection method, rather than a symbolic regression algorithm. The paper lacks comparison with other variable selection algorithms. In addition, in the noise robustness experiment, the paper only considered the presence of noise in the output variable y, which is inconsistent with reality. In practice, both input and output variables may have noise.

**Methods And Evaluation Criteria:**

Yes，The proposed methods and/or evaluation criteria (e.g., benchmark datasets) make sense for the problem or application at hand.

**Other Comments Or Suggestions:**

All comments and suggestions are provided above.

**Other Strengths And Weaknesses:**

The paper has done a great job, with standardized writing and clear discourse. However, there are the following suggestions for the author to further improve.
1.The paper constructed a new high-dimensional SR dataset for testing variable selection, which is a good work. However, the current constructed dataset is still too simple. It is possible to consider constructing some more difficult redundant variables, such as constructing new redundant variables like x3=x1+x2.
2. The paper only considered adding noise to the output variable y in the experiment, and it is recommended that the author also add noise to the input variable in the future.

**Questions For Authors:**

All comments and suggestions are provided above. Here, I would like to emphasize the two issues that concern me the most: first, the comparison with other variable selection methods, and second, adding noise to the input variables as well.

**Relation To Broader Scientific Literature:**

The key contributions of the paper are related to the correlation analysis and information theory in other literature. In previous studies, correlation coefficients and mutual information between variables were often used to determine whether variables were correlated.

**Theoretical Claims:**

The paper has no theoretical proof, and its contribution lies in the design of the algorithm. However, I think the author's emphasis on the importance of low FNR is correct.

---

> ### Author Rebuttal · Authors · 2025-03-31
>
> We appreciate the reviewers' thoughtful, constructive, and positive feedback on our work. We are glad that the motivation behind our approach was found to be clearly presented (pCrp, gjaN, pZx6), and that our focus on minimizing false negatives (FNs) in variable selection (VS) for symbolic regression (SR) was considered both important and well justified (pCrp, gjaN). We're pleased that reviewers found our method to be novel and effective (gjaN, GHAd, pZx6), and that the empirical evaluation was described as solid, convincing, and comprehensive (gjaN, GHAd, pZx6). We also appreciate the recognition of PAN+SR's novel direction to improve scalability in SR (pZx6), as well as our contribution to SR benchmarking and software development (gjaN).
>
> We'd like to take this opportunity to address your specific concerns and suggestions. We will include necessary revisions to clarify our work and strengthen the manuscript.
>
> ### Concern 1: Lack of comparison with other VS methods
> Contrary to the impression that no comparisons were made, our paper includes a **comprehensive evaluation of the proposed VS method (PAN) against 4 strong nonparametric VS methods**. These results are presented in **Appendix D.2** and referenced in **Section 4 paragraph 3**, where we note PAN consistently achieves the highest true positive rate (TPR) across all settings. We'll make this pointer more prominent in the revision to avoid confusion.
>
> Second, while the suggested papers share some surface-level similarities with our work and are worth citing, their focus is meaningfully different. Specifically, they use genetic programming (GP)—an SR method—as a tool for VS. In contrast, our work develops a **model-free VS** method that serves as a pre-screening step to improve the scalability and performance of SR methods, including but not limited to GP-based SR.
>
> Moreover, GP-based VS methods have several scalability and expressive power limitations. In addition to the number of input variables $p$, their effectiveness and tractability also depend heavily on:
>
> 1. The complexity and size of the operator set used for expression constructions
> 2. The maximum tree depth allowed
>
> As these grow, GP's search space **expands combinatorially**, making it less scalable and less suitable for high-dimensional problems. Furthermore, the reliance on a **pre-defined operator set** makes them inherently not model-free, while our proposed approach is designed to be **modular, scalable, and model-free**.
>
> Finally, we respectfully disagree with the characterization that "the contribution of the paper lies in proposing a variable selection algorithm." Our work goes beyond this—we address scaling SR to extreme dimensions, where the synergy between VS and SR is central. Our key message, supported by comprehensive experiments, is that a carefully designed model-free VS step consistently improves downstream SR across diverse methods.
>
> ### Concern 2: Performance under correlated and/or noisy predictors (also raised by gjaN and GHAd).
> We also appreciate your thoughtful suggestion regarding noise and correlation in predictors. While standard practice in SR benchmarks focuses on no/low noise in the output and independence in the predictors, we agree that noisy and/or correlated inputs represent a realistic and important challenge. That said, our current experimental design already introduces several challenges **beyond existing SR benchmarks**:
>
> 1. high dimensionality
> 2. irrelevant features
> 3. high output noises (8 levels)
> 4. low sample sizes (4 levels)
>
> leading to $8 \cdot 4 \cdot 100 \cdot 10=\textbf{32000}$ simulation settings. Although we're not able to test all suggested settings, they're not uncovered by our work. First, it's worth noting that PAN+SR already **performs well on the real-world datasets** (see Figure 1), where **27/35 (77%)** of datasets have at least 1 pair of predictors with **correlation >= 0.85**.
>
> In addition, **motivated by your comment**, we conducted a pilot experiment using the Friedman equation (also used in [3]) under a **high correlation structure**. Predictors $x_1,\ldots,x_p$ are drawn from a multivariate Uniform(0,1) distribution with an autocorrelation structure: $\rho_{ij} = 0.9^{|i-j|}$, and the responses were generated as
>
> $$y = 10\sin(\pi x_1x_2) + 20(x_3 – 0.5)^2 + 10x_4 + 5x_5 + \varepsilon, \qquad\varepsilon \sim N(0,\sigma^2).$$
>
> We fixed $n=1000$, $p=100$, and ran 100 trials per SNR level:
>
> | Metric | SNR=10 | SNR=5 | SNR=1 |
> |-|-|-|-|
> | TPR    | 100%   | 100%   | 99.8%  |
> | FPR    | 15.98% | 23.26% | 33.23% |
>
> Even so, **PAN consistently identified all 5 relevant predictors**, except for 1 run under SNR=1. We plan to include the discussion of correlated structure in the paper to help motivate future research focused on more realistic simulation settings.
>
> Thank you again for your time and constructive feedback. We appreciate the opportunity to clarify our work and believe these revisions will help strengthen the paper.

---

> > ### Comment · Reviewer_pCrp · 2025-04-03
> >
> > 1.In the comparative experiment in Appendix D.2, although the proposed algorithm has the highest TPR, its FPR is much higher than other algorithms, which may not be convincing.
> > 2.Regarding the issue of noise, what I mean is that in practice, noise exists in both the independent(input) and dependent(output) variables, not just in the dependent variable(output).
> > 3.One of my suggestion  is to construct some more difficult datasets, such as constructing x3=x1+x2, where x3 can be represented by x1 and x2, and therefore x3 can be deleted.

---

> > > ### Author Response · Authors · 2025-04-03
> > >
> > > Thank you for reading our Appendix and for the opportunity to clarify these points. We sincerely appreciate your constructive suggestion—it helped us strengthen the validation of PAN and enhance the realism of our simulation framework. **We hope these clarifications and additional experiments support a more favorable evaluation of our work.**
> > >
> > >
> > > ## Point 1. Asymmetric role of TPR and FPR in the context of SR
> > >
> > > First, we completely agree that in **conventional variable selection** (VS) problems, the trade-off between TPR and FPR is important, and a balance between the two is typically expected. However, our goal is to develop **VS methods specifically to support the scaling of SR methods** to large-$p$ datasets. This focus naturally leads to an asymmetric role of TPR and FPR. In particular, an effective VS method in our setting should prioritize maximizing TPR (ideally 100%), and only then aim to minimize FPR (which is secondary). This is because, in SR, the cost of excluding a relevant predictor (i.e., a FN) is much higher than that of including an irrelevant one (i.e., a FP). For example, if the true expression is $y = x_1 + x_2$ and $x_2$ is mistakenly excluded during pre-screening, **the correct expression becomes unrecoverable**. In contrast, if the selected set includes $x_1, x_2, x_3, x_4$, the correct expression remains accessible, and the irrelevant predictors can be ignored during symbolic expression search. See Lines 50-71 (left),119-128 (left), 148-152 (left), 251-260 (right), 355-384 (left), 405-419 (right) for related discussion.
> > >
> > > Our design choice of favoring FPs over FNs reflects a conservative and robust approach: it errs on the side of inclusion when uncertainty is high, ensuring that true signals are retained. In Appendix D.2, we show that PAN has the highest TPR (or the lowest FNR) across all settings, which is exactly the desirable behavior in SR pre-screening.
> > >
> > > Second, many SR algorithms incorporate implicit regularization mechanisms—such as penalizing complexity or preferring parsimonious expressions—that help filter out spurious variables during the model construction stage. Thus, even if some irrelevant predictors pass through the VS stage, they are unlikely to persist in the final expression, further reducing FPR.
> > >
> > > We hope this helps clarify the key distinction between our work and conventional VS problems, and highlights our unique focus driven by the challenges of extreme-scale SR.
> > >
> > > ## Points 2 & 3
> > >
> > > First, we fully agree that in real-world applications, noise can affect both the input (predictor) and output variables. In fact, this is reflected in the real-world datasets we evaluated, where PAN+SR demonstrates strong performance—suggesting robustness to such noise (and/or other scenarios not fully covered by our simulation) even without explicitly simulating it. We believe these real-data results offer a compelling and complementary benchmark alongside our simulations.
> > >
> > > Second, while our current simulation framework does not include noise in predictors, we have already incorporated four major challenges that are underrepresented in standard SR benchmarks: (1) high dimensionality, (2) presence of irrelevant predictors, (3) varying sample sizes (4 levels), and (4) multiple levels of output noise (8 levels), totaling 32,000 distinct settings. Extending this already comprehensive design to include input noise and/or redundancy for each of the 32,000 settings would significantly increase the computational burden and is beyond the scope of this study.
> > >
> > > However, motivated by your comment, we managed to extend the Friedman simulation used in our earlier response with **several new scenarios that directly address your Point 2 and Point 3**:
> > >
> > > 1. Baseline: Independent, noiseless, irreducible predictors
> > > 2. Noisy predictors (Point 2): Gaussian noise added to each predictor with noise variance equal to 1/5 of the signal variance
> > > 3. Duplicate predictor (Point 3): $x_6 = x_1+x_2$, where $x_1$ and $x_2$ are relevant predictors.
> > > 4. Correlated predictors: As in our earlier rebuttal, predictors follow an autoregressive structure with $\rho_{ij} = 0.9^{|i-j|}$
> > >
> > > All scenarios include additive output noise with a SNR of 10. Each setting was repeated 100 times, and the average performance is summarized below:
> > >
> > > | Scenario | TPR | FPR |
> > > |-|-|-|
> > > | Baseline       | 100% | 10.58% |
> > > | Noisy $X$      | 100% | 26.42% |
> > > | Duplicated $X$ | 100% | 11.11% |
> > > | Correlated $X$ | 100% | 15.98% |
> > >
> > > **PAN consistently achieves a 100% TPR**, showing robustness to (1) input noise, (2) redundancy via linear combinations, and (3) strong correlation structures in the predictors.
> > >
> > > We will include these new results in the final version of the paper, along with the corresponding code in our GitHub repository to ensure reproducibility. We sincerely thank you for this constructive suggestion, which has helped us further validate the robustness of PAN and improve the realism of our simulation framework.

---

### Official Review · Reviewer_gjaN · 2025-03-07

**Overall Recommendation:** 3

**Summary:**

This paper proposes a rank-clustering PAN strategy for screening for relevant features before running symbolic regression (SR) methods on very high dimensional data, where the goal is to minimize the false negative rate (avoiding missing important variables). The idea is to repeatedly run BART and use the ranking of feature importance measures to cluster relevant and irrelevant features. Then, only the features identified as relevant are fed into a symbolic regression algorithm. Through extensive simulation studies, this strategy is shown to be effective in improving the performance and reduce the running time of all popular downstream SR algorithms.

**Claims And Evidence:**

The claims for the benefits of the proposed method are mostly supported by empirical evidence. It would be nice to have some theoretical insights.

**Essential References Not Discussed:**

I haven't found such a case.

**Experimental Designs Or Analyses:**

I have checked the experimental setups, including data generating process, algorithms used, and their implementations. Existing experiments are solid, but it would further benefit from a more comprehensive design, which I elaborate in the "Questions" section.

**Methods And Evaluation Criteria:**

Yes.

**Other Comments Or Suggestions:**

- While the problem considered is well motivated and the argument on limiting FNR is solid, the proposed solution seems rather heuristic. It is not clear why the authors settle down to this specific solution. Some theoretical justifications will be helpful in establishing this method.
- The experiments are limited to i.i.d. covariates where the importance of features can be readily assessed by ranking and clustering. However, this kind of ranking strategy might be problematic when features are correlated. It will be helpful to extend the experiments.

**Other Strengths And Weaknesses:**

Please see my comments and questions.

**Questions For Authors:**

Similar to my comments,
- Is there theoretical justification for the method? Why do you pick this specific strategy?
- How do the methods work for correlated features?

**Relation To Broader Scientific Literature:**

The key contributions include:
- Formalize the goal of variable selection in SR (minimizing FNR) and its distinction from standard variable selection.
- Propose a strategy to conduct variable selection, which boosts existing methods, and may also inspire future development in this direction.
- Add to the software and benchmark development of the SR field.

**Theoretical Claims:**

This submission does not contain much theoretical aspects.

---

> ### Author Rebuttal · Authors · 2025-03-31
>
> We appreciate the reviewers' thoughtful, constructive, and positive feedback on our work. We are glad that the motivation behind our approach was found to be clearly presented (pCrp, gjaN, pZx6), and that our focus on minimizing false negatives (FNs) in variable selection (VS) for symbolic regression (SR) was considered both important and well justified (pCrp, gjaN). We're pleased that reviewers found our method to be novel and effective (gjaN, GHAd, pZx6), and that the empirical evaluation was described as solid, convincing, and comprehensive (gjaN, GHAd, pZx6). We also appreciate the recognition of PAN+SR's novel direction to improve scalability in SR (pZx6), as well as our contribution to SR benchmarking and software development (gjaN).
>
> We'd like to take this opportunity to address your specific concerns and suggestions. We will include necessary revisions to clarify our work and strengthen the manuscript.
>
> ### Concern 1: Motivation and theoretical justification for the proposed method (PAN)
> We agree that the motivation for PAN can be more clearly articulated. PAN is motivated by the observation that VIP rankings, unlike raw VIP values, tend to exhibit a clear **2-group structure** separating relevant from irrelevant predictors. This **2-group structure** is obscured in raw VIPs due to their bounded nature (summing to 1), whereas the ranking scale is wider and more interpretable. In Section 4 (paragraphs 5-8) and Appendix D.1, we show that under the uniform assumption, the expected mean rank $\bar{r}_{j\cdot}$ equals $(1+p_0)/2$ for relevant predictors and $(p_0+1+p)/2$ for irrelevant ones, naturally forming 2 clusters.
>
> This led us to **frame VS as a clustering problem** over mean VIP rankings $\bar{r}_{j\cdot}$. Although not included in the paper, we tested several clustering algorithms, including HAC, $k$-means, affinity propagation, Gaussian mixture model (GMM), spectral clustering, mean shift, DBSCAN, and BIRCH. We found that **HAC consistently outperformed others**. A key reason is its **robustness to class imbalance**, which is intrinsic to sparse regression problems where $p_0 \ll p$. In imbalanced data, large clusters can dominate centroid positions and overshadow the density signals of smaller clusters, making centroid-based and density-based methods less suitable for this task. In contrast, HAC starts with each point as its own cluster and merges based purely on pairwise distances. This allows small clusters (i.e., relevant predictors) to remain distinguishable.
>
> We plan to add an ablation study to compare the effect of different clustering algorithms on selection accuracy to justify our design choice further. We believe these additions will strengthen the motivation behind PAN and provide useful insights into the challenges posed by class imbalance in VS.
>
> ### Concern 2: Performance under correlated predictors (also raised by pCrp and GHAd)
> We appreciate your suggestion regarding correlation in the predictors. While the standard practice in SR benchmarks focuses on independence among the predictors, we agree that correlated inputs represent a realistic and important challenge. That said, our current experimental design already introduces several challenges **beyond existing SR benchmarks**:
>
> 1. high dimensionality
> 2. irrelevant features
> 3. high output noises (8 levels)
> 4. low sample sizes (4 levels)
>
> leading to $8 \cdot 4 \cdot 100 \cdot 10=\textbf{32000}$ simulation settings. Although we're not able to test all suggested settings, they are not uncovered by our work. First, it's worth noting that PAN+SR already **performs well on the real-world datasets** (see Figure 1), where **27/35 (77%)** of datasets have at least 1 pair of predictors with **correlation >= 0.85**.
>
> In addition, **motivated by your comment**, we conducted a pilot experiment using the Friedman equation under a **high correlation structure**. Predictors $x_1,\ldots,x_p$ are drawn from a multivariate Uniform(0,1) with an autocorrelation structure: $\rho_{ij} = 0.9^{|i-j|}$, and the responses were generated as
>
> $$y = 10\sin(\pi x_1x_2) + 20(x_3 – 0.5)^2 + 10x_4 + 5x_5 + \varepsilon, \qquad\varepsilon \sim N(0,\sigma^2).$$
>
> We fixed $n=1000$, $p=100$, and ran 100 trials per SNR level:
>
> | Metric | SNR=10 | SNR=5 | SNR=1 |
> |-|-|-|-|
> | TPR    | 100%   | 100%   | 99.8%  |
> | FPR    | 15.98% | 23.26% | 33.23% |
>
> Even so, **PAN consistently identified all 5 relevant predictors**, except for 1 run under SNR=1. We plan to include the discussion of correlated structure in the paper to help motivate future research focused on more realistic simulation settings.
>
> Thank you again for your time, constructive feedback, and thoughtful suggestions. We appreciate the opportunity to clarify our work and believe the revisions and additional discussions will help strengthen the paper.

---

### Official Review · Reviewer_GHAd · 2025-03-10

**Overall Recommendation:** 4

**Summary:**

The authors propose a feature selection preprocessing step to enhance the performance of symbolic regression algorithms. They introduce the method and evaluate its usage on SRBench across a number of algorithms and datasets.

**Claims And Evidence:**

The authors do a good job overall of making evidence-based claims. Evidence is convincing as it is over multiple algorithms, datasets, levels of noise, across metrics, and in comparison to other feature selection strategies (appendix).

**Essential References Not Discussed:**

There are some SR-based feature selection research papers that might be worth including but it is ancillary.

**Experimental Designs Or Analyses:**

Authors appear to follow recommended benchmarking practices. There aren't statistical tests for differences but the effect sizes are reported and laid out visually.

**Methods And Evaluation Criteria:**

Overall the methods make sense and the evaluation is commendable - the authors test their algorithm in combination with a number of SR algorithms and across real-world and synthetic datasets in a robust way. They also compare their feature selection strategy with others in terms of false positive/negative rates (of feature selection) with different levels of noise.

The only part of the methodology that didn't make total sense to me was the use of hierarchical agglomerative clustering to partition the feature space into two groups. It seems like overkill, and it isn't fully motivated. if you're just clustering the feature ranks over many trials, can't you use a much simpler threshold finder than HAC?

**Other Comments Or Suggestions:**

- redefine PAN outside of the abstract
- the authors motivate the problem by mentioning SR is NP-hard, but they might as well also note that feature selection is hard as well (http://www.jmlr.org/proceedings/papers/v40/Foster15.pdf)

**Other Strengths And Weaknesses:**

Overall I thought the paper was well done.

One weakness, in general, is that the strategy of preferring False Positives to false negatives may be less optimal on real-world data where there is a lot of cross-correlation, confounding, and redundancy in the dataset. It's a bit easier to optimize for that criteria on synthetic benchmarks of known physical systems than it might be for unknown systems. The comparison on real-world datasets mitigates some of this worry but it is worth mentioning.

**Questions For Authors:**

None apart from those mentioned thus far

**Relation To Broader Scientific Literature:**

Most work in SR develops a new method and then reports its benchmark results on SRBench in comparison to previous results. The authors instead evaluate a broadly applicable preprocessing strategy that synergizes with many methods.

The relevant literature is appropriately cited in symbolic regression, and reviewed in terms of feature selection as well.

**Theoretical Claims:**

There are no proofs

---

> ### Author Rebuttal · Authors · 2025-03-31
>
> We appreciate the reviewers' thoughtful, constructive, and positive feedback on our work. We are glad that the motivation behind our approach was found to be clearly presented (pCrp, gjaN, pZx6), and that our focus on minimizing false negatives (FNs) in variable selection (VS) for symbolic regression (SR) was considered both important and well justified (pCrp, gjaN). We're pleased that reviewers found our method to be novel and effective (gjaN, GHAd, pZx6), and that the empirical evaluation was described as solid, convincing, and comprehensive (gjaN, GHAd, pZx6). We also appreciate the recognition of PAN+SR's novel direction to improve scalability in SR (pZx6), as well as our contribution to SR benchmarking and software development (gjaN).
>
> We'd like to take this opportunity to address your specific concerns and suggestions. We will include necessary revisions to clarify our work and strengthen the manuscript.
>
> ### Concern 1: Motivation behind HAC
> We fully agree with you that the usage of HAC is not fully motivated, and further justification is warranted in the main paper. We chose to skip this motivation primarily due to the page limit. Although not included in the paper, we tested several clustering algorithms, including HAC, $k$-means, affinity propagation, Gaussian mixture model (GMM), spectral clustering, mean shift, DBSCAN, and BIRCH. We found that **HAC consistently outperformed others**. A key reason for choosing HAC is its **robustness to class imbalance**, which is intrinsic to sparse regression problems where the number of relevant predictors ($p_0$) is much smaller than the number of irrelevant ones ($p-p_0$). In imbalanced data, large clusters can dominate centroid positions and overshadow the density signals of smaller clusters, making centroid-based and density-based methods less suitable for this task. In contrast, HAC starts with each point as its own cluster and merges based purely on pairwise distances. This allows small clusters (i.e., relevant predictors) to remain distinguishable.
>
> We appreciate you highlighting this gap and will revise the manuscript to include a **dedicated explanation of our motivation and intuition in the method section**. We also plan to **add an ablation study** to compare the effect of different clustering algorithms on selection accuracy to justify our design choice further. We believe these additions will strengthen the motivation behind PAN and provide useful insights into the challenges posed by class imbalance in VS.
>
> ### Concern 2: Performance under correlated structure (by pCrp and gjaN)
> We appreciate your insightful comment on the role of correlated, confounding, and redundant variables in real-world settings. While our strategy of favoring false positives (FPs) over false negatives (FNs) may lead to more FPs in the presence of correlated predictors, we view this as a desired property. When uncertainty is high, it is preferable to retain a broader pool of potentially relevant features rather than risk excluding true signals. Furthermore, SR algorithms typically possess implicit regularization, which helps to further filter out irrelevant variables during model construction.
>
> As you rightly noted, PAN+SR already **performs well on the real-world datasets**, where **27/35 (77%)** of datasets have at least 1 pair of predictors with **correlation >= 0.85**.
>
> In addition, **motivated by your comment**, we conducted a pilot experiment using the Friedman equation under a **high correlation structure**. Predictors $x_1,\ldots,x_p$ are drawn from a multivariate Uniform(0,1) distribution with an autocorrelation structure: $\rho_{ij} = 0.9^{|i-j|}$, and the responses were generated as
>
> $$y = 10\sin(\pi x_1x_2) + 20(x_3 – 0.5)^2 + 10x_4 + 5x_5 + \varepsilon, \qquad\varepsilon \sim N(0,\sigma^2).$$
>
> We fixed $n=1000$, $p=100$, and ran 100 trials per SNR level:
>
> | Metric | SNR=10 | SNR=5 | SNR=1 |
> |-|-|-|-|
> | TPR    | 100%   | 100%   | 99.8%  |
> | FPR    | 15.98% | 23.26% | 33.23% |
>
> Even so, **PAN consistently identified all 5 relevant predictors**, except for 1 run under SNR=1. We plan to include the discussion of correlated structure in the paper to help motivate future research focused on more realistic simulation settings.
>
> We are also more than happy to incorporate your suggestions on redefining PAN outside of the abstract and to motivate the difficulty of variable selection using the provided reference. Thank you again for your time, constructive feedback, and thoughtful suggestions. We appreciate the opportunity to clarify our work and believe the revisions and additional discussions will help strengthen the paper.

---

### Official Review · Reviewer_pZx6 · 2025-03-11

**Overall Recommendation:** 4

**Summary:**

The authors are interested in the problem of discovering mathematical equations from raw data. One of the largest bottleneck for SR methods is that it's extremely hard to scale the equation search to > 10 variables. This is because each additional variable considered combinatorially increases the search space which makes the search less efficient. The authors hypothesize that, given a large number of variables, we can analyze certain statistical properties to cluster the variables based on their relevance and prune the irrelevant variables, which should considerably increase the search efficiency. Towards this extend, the authors propose PAN-SR, a framework which modifies the BART variable selection method to select a set of relevant variables to run an off-the-shelf SR algorithm on.

The authors demonstrate that, for almost all models on SRBench, preprocessing the input with PAN-SR improves performance (although there is a higher delta improvement for lower performing models compared to higher performing models).

**Claims And Evidence:**

Claim: PAN+SR is a general purpose algorithm to increase SR scalability.

Comments: I'm not 100% convinced that PAN+SR "solves" the scalability problem but the authors present a novel direction of improvement for SR scalability which is pretty exciting. Previous work has generally handled this scalability challenge by (1) inducing programs with neural networks (which exposes an out of distribution problem) `[1, 2]` and (2) by using LLMs to induce programs  `[3, 4, 5]`  which are expensive to run. PAN+SR circumvents this problem by pre-selecting the variables but I'm not sure if a clean variable clustering exists in extremely noisy settings. Regardless, PAN+SR provides empirical justification that the model scales better than baseline models.


Claim: PAN is the most performant pre-screening strategy.
(Not really an explicit claim but an implicit one)

Comments: I generally agree that the algorithm is well motivated but it's not completely certain whether the increased performance is a result of pre-screening in general or pre-screening specifically with PAN+SR. Some additional experiments using other variable selection methods (e.g: BART) would be extremely insightful here.


`[1]`: https://proceedings.mlr.press/v139/biggio21a/biggio21a.pdf

`[2]`: https://github.com/deep-symbolic-mathematics/TPSR

`[3]`: https://arxiv.org/abs/2404.18400

`[4]`: https://arxiv.org/abs/2409.09359

`[5]`: https://ai-2-ase.github.io/papers/52_InceptionSR_AAAI_25.pdf

**Essential References Not Discussed:**

.

**Experimental Designs Or Analyses:**

The methodology is sufficiently sound. The authors utilize and describe the evaluation setup proposed in SRBench.

**Methods And Evaluation Criteria:**

The authors use a modified version of SRBench to simulate sampling equations with noisy irrelevant variables. These variables are drawn from the same data distribution to increase the task hardness. Overall, I found the evaluation criteria to be well motivated.

One small nitpick: The data for each variable in SRBench (ground-truth) is sampled from a normally distributed random variable. Would PAN+SR's performance suffer if the empirical data is not normally distributed?  Specifically, we know that physical laws tend to have very diverse data distributions `[6]`. e.g.: Newton's law and Coulomb's law have very similar equation sketches but the scale at which they operate is extremely different. Specifically, an additional experiment of PAN+SR's performance on `[6]` would be extremely helpful!

Overall, I believe this paper will lead to great discussions at the conference and am in favor of **accepting** this paper. PAN+SR presents a new and refreshing direction for scaling SR methods by pre-screening the variable set using statistical heuristics that is empirically validated.


`[6]`: https://arxiv.org/abs/2206.10540

**Other Comments Or Suggestions:**

.

**Other Strengths And Weaknesses:**

I generally found the paper to be polished and easy to read. I think a bit more time can be devoted (in the appendix maybe) to go through how the pre-screening strategy would work on a small example, but otherwise this was a pretty interesting read.

**Questions For Authors:**

.

**Relation To Broader Scientific Literature:**

.

**Theoretical Claims:**

.

---

> ### Author Rebuttal · Authors · 2025-03-31
>
> We appreciate the reviewers' thoughtful, constructive, and positive feedback on our work. We are glad that the motivation behind our approach was found to be clearly presented (pCrp, gjaN, pZx6), and that our focus on minimizing false negatives (FNs) in variable selection (VS) for symbolic regression (SR) was considered both important and well justified (pCrp, gjaN). We're pleased that reviewers found our method to be novel and effective (gjaN, GHAd, pZx6), and that the empirical evaluation was described as solid, convincing, and comprehensive (gjaN, GHAd, pZx6). We also appreciate the recognition of PAN+SR's novel direction to improve scalability in SR (pZx6), as well as our contribution to SR benchmarking and software development (gjaN).
>
> We'd like to take this opportunity to address your specific concerns and suggestions. We will include necessary revisions to clarify our work and strengthen the manuscript.
>
> ### Concern 1: Scalability challenge being solved by PAN+SR
> We agree that PAN+SR does not completely resolve the scalability challenge, but we believe it takes a significant step forward. In addition to the performance improvements presented, we'd like to share further concrete evidence, which will be included in the final paper for clarity.
>
> For context, the average values of $p$ and $p_0$ in the Feynman datasets are 186.15 and 3.65, respectively. On average, PAN reduces $p$ to 5.23 (97% reduction) in the best-case scenario (no noise), and to 63.1 (66% reduction) in the worst-case scenario (SNR=0.5). Furthermore, scalable SR methods such as TPSR [2] and DySymNet also benefit from PAN+SR, as shown in Figures 1 & 2. We chose not to include NeSymReS [1] as a baseline because TPSR [2] already employs a pre-trained NeSymReS backbone in our implementation.
>
> ### Concern 2: Existence of a clear separation under noisy settings
> As you rightly pointed out, separating signal from noise becomes harder in extremely noisy settings (Figures 5 & 6 in Appendix D.1). Nonetheless, relevant predictors still tend to cluster around the low-mean cluster, which explains the consistently high true positive rate (TPR) despite heavy noise (Figure 7 in Appendix D.2).
>
> ### Concern 3: PAN's role in improving performance
> In Appendix D.2, we compared PAN against 4 other nonparametric VS methods on the high-dimensional Feynman database. As shown in Figures 7 and 8 of Appendix D.2, PAN consistently achieves the highest TPR across all settings, albeit at the cost of frequent false positives. Since identifying TPs is more critical in the SR pre-screening context, we believe PAN offers the most practical, effective, and safe solution.
>
> ### Concern 4: PAN's robustness to different sampling distributions for the predictors
> Your observation about the distributional assumptions in SRBench and the suggestion to consider [6] are also much appreciated. Indeed, each variable in the Feynman dataset is sampled from a uniform distribution, a common design choice in empirical studies aimed at ensuring even coverage of the input space to support the generalizability and robustness of the study. As [6] highlights, however, the sampling range of variables can influence SR performance. We agree with many points raised in [6] and find it to be a valuable reference for the related work section.
>
> However, we do not expect sampling range to impact the pre-screening performance of PAN+SR. This is because **tree-based methods like BART are invariant to monotonic transformations of the input features**. Thus, the pre-screening result should remain robust to variation in the sampling range.
>
> Additionally, while [6] enhances the Feynman dataset by adding 1-3 irrelevant variables, our experimental setup considers a **substantially more challenging high-dimensional regime**, adding $50p_0$ irrelevant variables (ranging **from 100 to 450**). We also note that while the original treatment of constants and integer-valued variables as real-valued variables does violate their physical meanings—as [6] rightly points out—this choice inadvertently increases the difficulty of the task, thereby providing a more stringent test for both pre-screening and SR modeling. For these reasons, we believe additional experiments based on [6] would not significantly change the conclusion of our study, though we acknowledge its importance and will include it in the discussion of future directions.
>
> However, motivated by other reviewers' comments, we will
>
> 1. add an **ablation study to compare the effect of different clustering algorithms**, and
> 2. include a **high correlation** experiment using the Friedman equation. See response to gjaN for details.
>
> We believe these additions will strengthen the motivation behind PAN and broaden our experimental evaluation.
>
> Thank you again for your time, constructive feedback, and thoughtful suggestions. We appreciate the opportunity to clarify our work and believe the revisions and additional discussions will help strengthen the paper.

---

> > ### Comment · Reviewer_pZx6 · 2025-04-02
> >
> > Thank you for the additional clarification. I'll be maintaining my current score. Great work!

---

### Decision · Program_Chairs · 2025-05-01

**Decision:**

Accept (poster)

**Comment:**

Overall, the reviewers showed positive impressions for this work and offered 1 accept and 3 weak accepts.

Many of the reviewers valued the proposed method and evaluation results that support the benefits of using the method.
To the best of my knowledge, it is rare to see discussions focused on feature selection in symbolic regression tasks as it's usually part of proposed SR methods. The proposed method is method agnostic as it's a preprocessing approach, and I believe such discussions should be very important for the community.

While I support accepting this work for ICML 2025, I want to make the following suggestions (2 and 3 are based on our internal discussion):
1. Update references section. I found many of papers cited as arXiv preprints published at some venues
2. Add an explicit section in the appendix that explains the dataset construction and sampling process, especially for irrelevant data e.g., how exactly the irrelevant features were generated such as range of values, distributions, whether or not the range is overlapped with those of the original features
3. In the camera-ready or future studies, consider using SRSD datasets (dummy variables ver.) for evaluations or at least follow their procedure to generate dummy (irrelevant) variables as it's more challenging (much larger range and independent from a given equation)